# Distractibility and impulsivity neural states are distinct from selective attention and modulate the implementation of spatial attention

J. L. Amengual[1,4] ✉, F. Di Bello[1,2,4], S. Ben Hadj Hassen[1,3] &
Suliann Ben Hamed [1] ✉

In the context of visual attention, it has been classically assumed that missing the response to a target or erroneously selecting a distractor occurs as a consequence of the (miss)allocation of attention in space. In the present paper, we challenge this view and provide evidence that, in addition to encoding spatial attention, prefrontal neurons also encode a distractibility-to-impulsivity state. Using supervised dimensionality reduction techniques in prefrontal neuronal recordings in monkeys, we identify two partially over-lapping neuronal subpopulations associated either with the focus of attention or overt behaviour. The degree of overlap accounts for the behavioral gain associated with the good allocation of attention. We further describe the neural variability accounting for distractibility-to-impulsivity behaviour by a two dimensional state associated with optimality in task and responsiveness. Overall, we thus show that behavioral performance arises from the integration of task-specific neuronal processes and pre-existing neuronal states describing task-independent behavioral states.

During our daily life, we are exposed to a myriad of situations during which we need to select and process different kinds of sensory events and to act accordingly. For example, we have to pay attention to the traffic light in order to start or stop our car or focus on a unique conversation while avoiding to listen to simultaneous irrelevant conversations. Attention plays a critical role in all of these situations, implementing the selection of the sensory cues that are relevant to our ongoing purposes[1–5] while suppressing the irrelevant information the response to which has to be inhibited[6]. However, other factors independent from the goals of the task are also expected to interfere, either enhancing or degrading behavioral performance, such as fatigue[7,8], motivation[6,9,10], the degree of liberal or conservative biases in response

performance[11] or intrinsic fluctuations of information coding of cognitive processes[12]. All of these factors arguably define "internal states" that highly influence perceptual outcomes under similar sensory conditions.

Prior electrophysiological studies in primates have focused on how these internal states organize and influence sensory processing and decision-making[11,13,14]. However, little is known on how such internal states are organized at the neuronal population level, how they functionally interact with attentional processes and how they account for different patterns of behavioral performance characterized by either distractibility (i.e., the subject's inattention to both task-relevant and task-irrelevant items) or impulsivity (i.e., the subjects'

[1]Institut des Sciences Cognitives Marc Jeannerod, CNRS UMR 5229, Université Claude Bernard Lyon 1, 67 Boulevard Pinel, 69675 Bron Cedex, France. [2]Department of Physiology and Pharmacology, Sapienza University of Rome, 0018 Rome, Italy. [3]Laboratory in Sensory Physiology, Otto Von Guericke University, 39120 Magdeburg, Germany. [4]These authors contributed equally: J. L. Amengual, F. Di Bello ✉e-mail: Julian.amengual@isc.cnrs.fr; benhamed@isc.cnrs.fr

propensity to respond to incoming stimuli irrespective of their relevance). This has a high relevance to the understanding of the neural bases of attention disorders such as attention deficit and hyperactivity disorder (ADHD), as recent studies describe that allocation of attention is not impaired in these patients[15], and other processes presumably related to the internal states described here could be affected.

The question we address in the present work is whether behavioral performance during an attentional task is (non-exclusively) (i) a direct consequence of a good or a miss allocation of spatial attention relative to task events[1,2,6,16–19], or (ii) determined by underlying internal states that can be assigned to specific functional neuronal states of the prefrontal cortex neural population responses[11]. We will focus on the macaque frontal eye fields (FEF), a brain region in the prefrontal cortex crucial to the voluntary control of attention[2,20]. Recent studies have shown that macaque monkeys showed a higher hit and false alarm rate on an attentional cued-target-detection task when the decoded position of the attentional spotlight was closer to the expected target or distractor location, respectively[6,16,21–23]. In addition, the level of statistical shared variability between neurons recorded in the FEF[24] measured within a time interval *prior* to attentional orienting by the cue (therefore, prior to any specific knowledge of the position of the stimulus to-be-attended) also predicted overt behavior of the monkey, showing lowest noise correlations in hit trials as compared to miss or false alarm trials[21]. In this respect, Nogueira and colleagues[25], show that two statistical features of the neural populations accounted for the amount of encoded information and behavioral outcomes,

namely the degree of attentional coding and the population covariability along this attentional coding axis. All this, taken together, strongly suggests that inappropriate behavior is only partially driven by the quality of the orientation of spatial attention. We thus hypothesize that the influence of other activity[26,27] independent of spatial attention orientation plays a critical role in behavioral performance.

To address this question, we trained two macaque monkeys to perform a 100% validity-cued visual attention task in the presence of distractors. We recorded multi-unit neuronal activity (MUA) from multiple recording sites in the FEF. Using machine learning techniques, we decoded the (x,y) position of the locus of attention at a high spatial and temporal resolution before target onset, which allowed us to classify trials based on how well attention was oriented according to task instructions. In agreement with previous studies[16,21], this metric of the accuracy of spatial attention orientation highly predicts overt behavioral performance in the task. However, it does not fully account for behavioral performance. Indeed, a substantial proportion of trials in which decoding indicated an attentional spotlight located close to the expected target position were not successful and the target was missed. Likewise, trials with poor decoded attention orienting could still end up as correct trials. We thus predicted that internal states associated with specific neuronal population states can be precisely identified and can either enhance or interfere with spatial attention processes. To prove our prediction, we used demixed principal component analysis (dPCA)[28,29], a dimensionality reduction technique that allows associating the variability explained by each component with specific task- or behavior-related parameters. We thus divided our trials based on the position of the decoded attentional spotlight with respect to the target position (target-to-attention distance, TA; Small TA, Medium TA, Large TA) and the behavioral performance (hit trials, miss trials, and false alarm trials), and we extracted demixed components the variance of which was associated with either to TA or behavioral performance. As predicted, we identify components in the neural population that specifically encode attention and upcoming behavior (hit/miss), respectively, and we show that the information for each parameter was encoded in overlapping neuronal populations. Importantly, we find that the smaller the overlap between these neural subpopulations, the higher the behavioral gain associated with an effective attention orientation, i.e., the smaller the interference between the internal states associated with behavioral outcome and visual attentional processes. In addition, when focusing on the neuronal variability accounting for the behavioral outcome, we identify specific neuronal subpopulations characterizing a two-dimensional internal state associated with different levels of distractibility and impulsivity in the responses. One of the state's dimensions defines a continuum of coding between distractibility (absence of response), optimal response, and impulsivity (inappropriate response), possibly suggesting an association with proactive inhibition and response threshold adjustment[11]. A second dimension singles out optimal response (hits) from misses and false alarms along a U-shaped relationship. Consistent with this finding, prior literature has reported evidence suggesting that such behavioral responses might be linked with different activity regimes described in the Locus Coeruleus[30]. Therefore, activity in this dimension might correspond to the neural signature of the activity of the noradrenergic system. Overall, this work provides direct evidence for a functional dissociation between spatial attention and the control of behavioral optimality, proposing a different framework for the interpretation of ADHD symptoms and associated neuropharmacological therapeutic approaches.

## Results and Discussion

### Attentional-orienting is only partly predictive of behavior

We recorded MUA from two 24-contact recording probes that were implanted in two macaque monkeys bilaterally in the FEF, on the anterior bank of the arcuate sulcus, during the execution of a

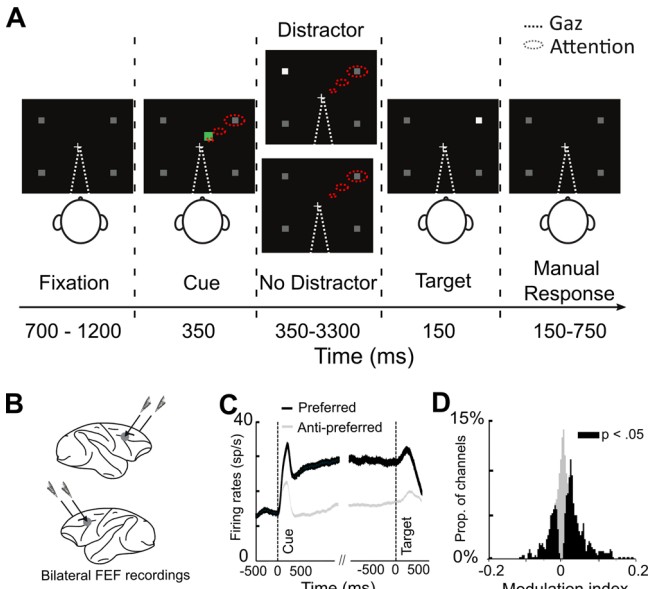

**Fig. 1 | Methods. A** 100 % validity cued target-detection task with distractors. To initiate the trial, monkeys had to hold a bar with the hand and fixate their gaze on a central cross on the screen. Monkeys received liquid reward for releasing the bar 150–750 ms after target presentation onset. Target location was indicated by a cue (green square, second screen). Monkeys had to ignore any un-cued event (distractors). Monkeys were instructed to keep their gaze on the fixation point (white dashed lines), therefore they had to detect the stimuli using selective spatial attention (red dashed lines). **B** On each session, one 24-contact recording probe was placed in right FEF (top) and left FEF (bottom). **C** Single MUA mean ( ± s.e.) associated to when cue is orienting towards the preferred (black) or the anti-preferred (gray) spatial location, during the cue-to-target interval. **D** Distribution of attention modulation index (Preferred-Anti-preferred)/(Preferred + Anti-Preferred), computed over 200 ms before target onset across all MUAs of all sessions. Black histogram corresponds to channels in which the neuronal activity during this time interval was significantly different between the preferred and the anti-preferred spatial attention responses (Two-sided Wilcoxon signed-rank test, $p < 0.05$: black, significant difference).

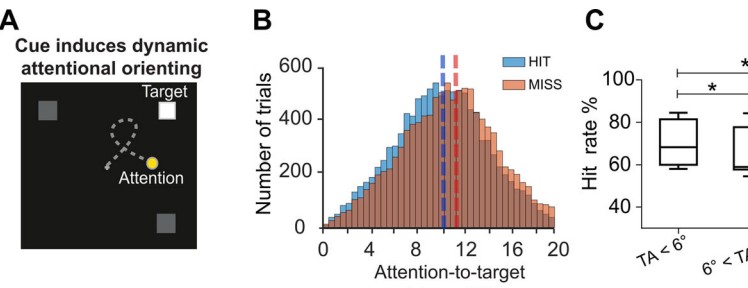

**Fig. 2 | Impact of attentional locus on behaviour. A** Attentional orientation: The presentation of the spatial cue instructs the monkeys to orient their attention towards the expected target location (white square). High-resolution spatial decoding of the position of the attentional locus allows to track the position of the attentional spotlight (yellow dot) on the screen prior the target onset (gray trajectory). **B** Histogram of the distribution of the number of hit and miss trials as a function of TA bins (after equalizing the number of hit and miss trials pear each TA bin). **C** Box plot showing the distribution of overall monkeys' accuracy in target detection (hit/(hit+miss)) as a function of the target to attention distance (TA-step = 6 degrees). Lower and upper box boundaries reflect the 90th and 10th percentiles, respectively, line inside box reflects the median and lower and upper error lines show the min and max value of each distribution, respectively (N = 18 sessions, two-tailed Wilcoxon signed-rank test, *p = 0.02, **p < 0.01).

visually cued target-detection task (Fig. 1A, B). Similar to what is described in previous studies[1,2,6,21–23], these recorded neurons show enhanced responses when attention is oriented towards their preferred receptive field (RF) location relative to when it is oriented towards the least preferred spatial location (Fig. 1C). The majority of neurons are significantly modulated depending on whether attention is oriented towards their preferred or non-preferred receptive field (Fig. 1D). This dataset has been used in prior studies from our laboratory[22,23].

Recent studies demonstrate that, on classical spatial attention tasks as used here, while attention is on average oriented to the cued location, it is actually intrinsically dynamic, exploring space rhythmically both across and within trials[6,23,31]. Here, we use machine learning applied to independent session recordings in order to assess the spatial locus of the attentional spotlight from FEF neuronal ensembles in time on independent trials (see material and methods, Fig. 2A and[6,21–23]. This decoded attentional spotlight is thus used as a proxy of the locus of attention. This allows us to measure, at the single-trial level, the distance of the decoded spotlight to the expected target position (Target-Attention distance; TA). This measure is a proxy of the focus of attention. Corroborating our previous observations[6,21–23], we find that the distribution of TA distance varied as a function of the perceptual behavioral outcome. Specifically, after equalizing the number of hit and miss trials per each recording session, the distribution of TA in misses is significantly shifted towards larger TA values with respect to hits (median hits: 10.12°, IQR: 5.5; median misses: 11.02°, IQR: 5.7, two-sided Kolmogorov−Smirnov test, D = 0.96, p = 0, Fig. 2B). Overall, the smaller the TA, the higher the hit rate (Friedman test X$^2$ (17) = 32.44, p = 1e-15, Fig. 2C). Trials with short TA (TA < 6°) are associated with significantly higher hit rates than when attention is located at intermediate locations (6° < TA < 12° · two-tailed non parametric Wilcoxon signed-rank test Z = 2.1, p = 0.02) or far away from the expected target location (12° < TA < 18° · two-tailed non parametric Wilcoxon signed-rank test, Z = 2.8, p = 0.004). The hit rates for these last two TA trial categories are also different from each other (two-tailed non parametric Wilcoxon signed-rank test, Z = 2.3, p = 0.018). Importantly, although TA has a strong effect on hit rates, this parameter does not fully account for behavioral outcome. Indeed, over ~20% of trials with short TA were misses, and up to 65% of trials with long TA were hits (Fig. 2B, see ref. 16). This indicates that the behavioral accuracy is not uniquely explained by the attentional focus and raises the possibility that other ongoing cognitive processes are engaged in parallel and impacting performance concurrently with the reallocation of attention.

## Prefrontal neurons are modulated both by the attentional focus and upcoming behavioral accuracy

During spatial attention tasks, neuronal FEF responses are classically shown to differ between correct (hit) and error (miss) trials[1,6,18,19,32,33]. This is often interpreted as an indication that on error trials, attention is not properly oriented to the instructed location. Here, for all recorded neuronal responses, we pooled trials based on both the TA category (close TA vs medium TA vs long TA, see the previous subsection) as well as based on behavioral accuracy (hit vs. miss). This TA measure is a different metric from attention orientation towards the preferred versus anti-preferred location on the screen, as it reflects how close the attentional spotlight is to the expected target location, irrespective of the actual target position (that is to say, the attentional focus). Figure 3A shows the signal of a neuron that is tuned to behavioral accuracy, with an overall higher spiking rate for hits than for misses (Fig. 3A, blue vs green shades). In each of the hit and miss trials, the neuron's activity is also tuned to the attentional focus, its activity being higher when attention is closer to the target (Fig. 3A, dark to light shades). As a result, the firing rate of this neuron cannot by itself predict how close the attentional spotlight is to the target and whether the monkey is going to produce a hit or a miss. For example, the neuron signals with the same level of firing rate a medium TA on upcoming correct trials and a close TA on upcoming missed trials. Figure 3B represents a second neuron showing the same properties, except for the fact that its activity is lower when attention is closer to as compared to when it is far away from the target (Fig. 3B, dark to light shades), both in hits (blue shades) and in misses (green shades).

At the population level, the closer the attentional spotlight was to the expected target location (i.e. prior to target presentation), the higher the spiking rate (Fig. 3C). Spiking rates were also significantly higher on hit trials than on miss trials (Fig. 3D). Thirty-five percent (35.8%) of recorded neuronal signals were significantly modulated by TA (181/505), and the majority of these signals (66.8%, 121/181) showed higher activity in trials where attention was closest to the target (−400 to −100 ms before the target onset, Fig. 3E). Independently, the response of 18.85% of the MUAs (95/505) is significantly different between hits and misses, 75.8% (72/95) of the MUAs having higher spiking rates on hits (−400 to −100 ms before target onset, Fig. 3F). As a result, the distribution of the modulation index based on the TA estimated prior to target presentation is biased towards positive values (Median = 0.046, IQR: 0.12, Fig. 3E), as well as the distribution of the behavioral outcome modulation index in the same time interval (Median = 0.026, IQR: 0.031, Fig. 3F).

Overall, this indicates that FEF cells encode, just prior to target onset, both how close attention is oriented to the expected target and whether the monkey is going to succeed on the trial or not. In order to

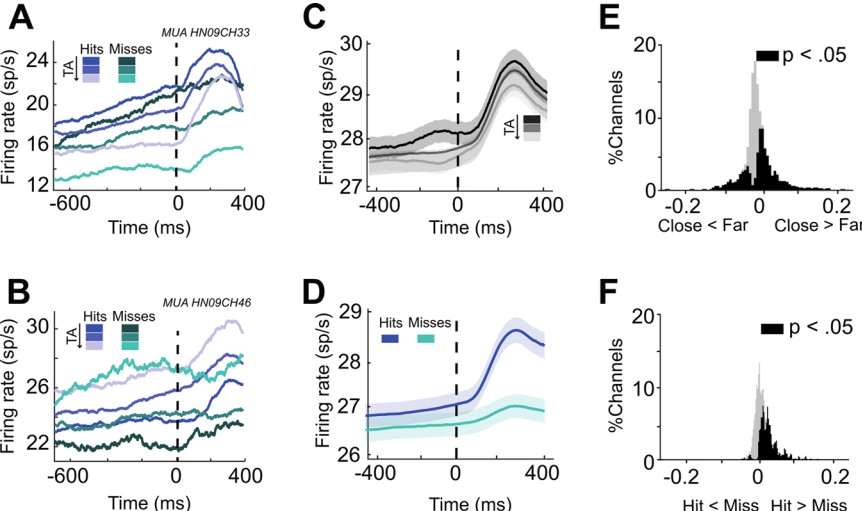

**Fig. 3 | MUA activity is modulated by the distance of spatial attention to the expected target and behavioral outcome. A, B** Single unit MUA activity pooled by TA (darker tones corresponding to smaller TA values) and upcoming target behaviour (hit trials blue, miss trials green) locked to the target onset (dashed vertical line). **C, D** Averaged MUA activity pooled by TA (darker tones corresponding to smaller TA values) and upcoming target behaviour (hit trials blue, miss trials green) averaged across all task-selective MUAs of all sessions. Activity is locked to target

onset. Shadowed bands indicate the s.e. (N = 596 task-related channels). **E, F** Distribution of the modulation index based on TA (**E**) and behavioral outcome (**F**) across all task-selective MUAs of all sessions (N = 596 task-related channels). Black histogram corresponds to channels in which the neural activity was significantly different between the two classes (TA: Close vs Far; Upcoming target behaviour: hit vs miss, non-parametric two-sided Wilcoxon signed-rank test, p < 0.05). Gray color indicates channels showing no significant difference.

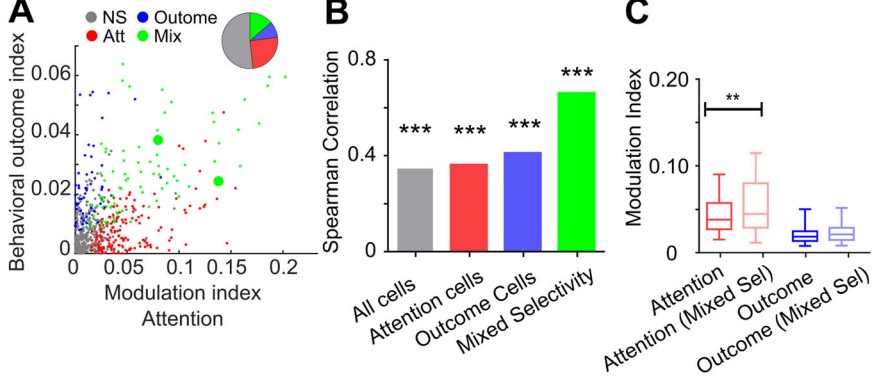

**Fig. 4 | Attentional and behavioral outcome mixed selectivity in the FEF cells. A** Scatter plot showing attentional (x-axis) and behavioral outcome (y-axis) modulation indices (absolute value) for all recorded task-related neurons (N = 596) during the time interval −400 to −100 ms before target onset. Neurons are classified based on the significant TA (Att, red; N = 150), behavioral outcome (Outcome, blue; N = 55) or mixed (Mix, green; N = 82) tuning. Neurons corresponding to Fig. 3A, B are indicated by larger symbols. Pie chart shows the proportion of neurons for each type of selectivity. **B** Bar plot indicating the spearman correlation coefficient between attention and behavioral outcome modulation indices (absolute value) for all task-related neurons (gray, N = 596), pure TA (red, N = 150),

behavioral outcome (blue, N = 55) selective neurons, and mixed selective (green, N = 82) neurons. Asterisks indicates the significance of the correlation (Spearman Correlation, p < 0.001 for all comparisons). **C** Box-plot of the distribution of modulation indices of TA (red shades) and behavioral outcome (blue shades) for mixed (light shades) and unmixed (dark shades) selectivity cells. Lower and upper box boundaries reflect the 90th and 10th percentiles, respectively, line inside box reflects the median and the lower and upper error lines show the min and max value of each distribution, respectively (** two-sided Wilcoxon rank-signed test p = 0.005).

better characterize how this is organized at the single-cell level, we compare the absolute value of the behavioral outcome and attentional modulation indices of each individual cell. Figure 4A shows that neurons fall along a continuum of combinations of attentional and behavioral outcome coding strength, with neurons showing strong attentional coding but weak behavioral outcome coding (and vice versa), as well as, neurons encoding strongly both parameters simultaneously. Based on the statistical significance of each of their individual attentional and behavioral outcome modulation indices, 19.1% of the neurons showing significant coding strength are found to uniquely encode behavioral outcome, 52.2% uniquely encode attention orientation, and 27.5% show mixed selectivity to both parameters (Fig. 4A). Globally, we find a significant correlation between behavioral outcome

and attentional modulation indices across all recorded neurons (Fig. 4B, Spearman correlation, $\rho = 0.35$, $p = 1e{-}27$). This correlation still held true when considering only attention modulated cells (Spearman correlation, $\rho = 0.36$, $p = 0.0079$), or cells modulated by only behavioral outcome (Spearman correlation, $\rho = 0.41$, $p < 0.001$). This thus indicates that there is a positive relationship in how both information (attention and behavioral outcome) are encoded at the single-neuron level. Interestingly, this correlation is stronger when only mixed selectivity neurons are considered (Spearman correlation, $\rho = 0.63$, $p = 1e{-}12$), indicating a stronger functional relationship between these neurons and overt behavior. In the prefrontal cortex, mixed selectivity neurons coding for both spatial attention and perception are shown to have higher attentional modulation indices than cells coding only for

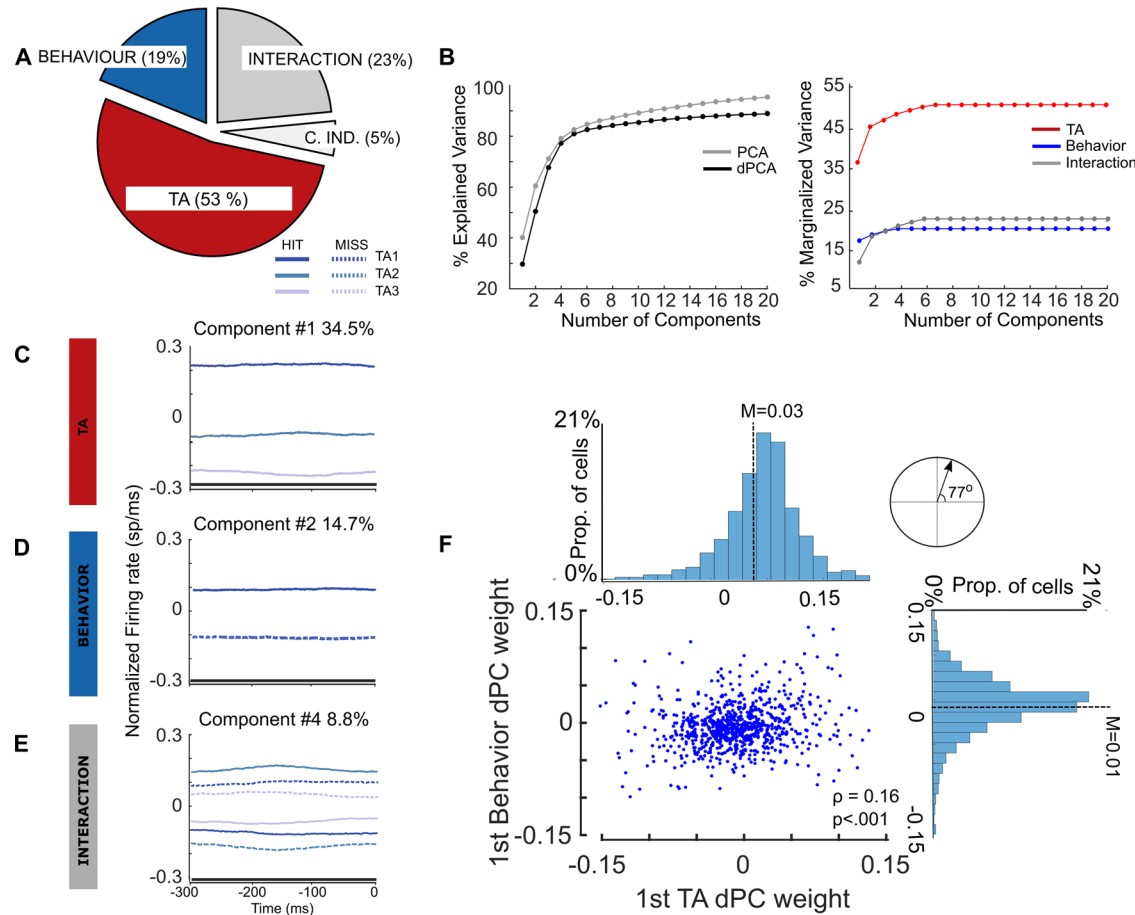

**Fig. 5 | Demixed PCA unmixes attention and behavioral outcome-related variance. A** Pie chart shows how the total signal variance is split among parameters: Attention (red), behavioral outcome (blue), Interaction between attention and behavior (dark gray), and the condition independent variance (light gray). (**B**, Left) Cumulative variance explained by PCA (gray) and dPCA (black). Demixed PCA explains almost the same amount of variance as PCA. (**B**, Right) Cumulative demixed variance specific for each marginalization. **C−E** Demixed principal component. In each plot, the full firing rates from −300 ms to 0 ms from target onset are projected onto the respective dPCA decoder axis (attention, behavioral outcome and interaction), as a function of trial type categories (based on TA distance and behavioral outcome) so that there are six lines corresponding with six conditions

(see legend). Thick black lines show time intervals during which the respective task parameters can be reliably extracted from single-trial activity (as assessed against a 95% C.I., permutation test). **F** For each neuron, we use the first attention- and behavioral outcome-related demixed PCs to plot its location on the plane defined by these two components. These components present a weight distribution that tends to be centered and equally distributed around zero (cf. respective histograms, $N = 848$ neurons). The scatter plot shows the relationship between the neurons' weights in the attention and behavioral outcome demixed component. This correlation is significant ($N = 848$ electrodes, Spearman correlation, $\rho = 0.16$, $p = 3\mathrm{e}{-4}$). The dot product between these components indicates that these components were non-orthogonal (77 degrees).

spatial attention, while perception-related modulation indices did not vary between mixed selectivity and pure perception neurons[34]. Similarly, we found that attentional modulation indices were significantly higher in mixed selectivity cells than in pure attentional cells (Fig. 4C, two-sided Wilcoxon rank-signed test $Z = 2.8$, $p = 0.0048$). In contrast, behavioral modulation indices did not vary between mixed selectivity neurons and pure behavioral outcome neurons (Fig. 4C, two-sided Wilcoxon rank-signed test $Z = 1.37$, $p = 0.11$). In summary, the FEF thus encodes spatial attention and behavioral outcome through mixed selectivity neurons. These mixed selectivity neurons[35–37] have higher attentional modulation indices and better account for attention than the unmixed selectivity FEF neurons.

**Both distance of attention to expected target and overt behavioral outcome account for prefrontal neuronal variability**

At the neural level, a subpopulation of neurons is thus modulated differently by the attentional focus and the upcoming behavior. This finding suggests the coexistence of at least two different processes in the neural activity of the FEF just prior to target presentation. In the previous section, we showed that behavioral and attentional focus coding was partially mixed in a subpopulation of recorded FEF

cells[35–37]. In order to study the interplay between neuronal subpopulations encoding attentional focus and predicting behavioral outcomes, we applied a dimensionality reduction to our data. Dimensionality reduction methods are very useful to project the neuronal population data from the high dimensional space onto a new low dimensional manifold such that some of its dimensions (or principal components) maximally account for a given source of neuronal variability. We applied a principal component analysis (PCA) to FEF neuronal responses pooled along TA categories and correct and incorrect trials. When independently projected onto the principal components, the neural responses for hits and misses, as well as for close, medium, and far TA distances, we identify a first component that mostly accounts for the upcoming behavioral outcomes (PC1) and a second and a third component that substantially account for attention to target distance (PC2 and PC3, Supplementary Fig. 1). Information related to the upcoming behavioral outcome and TA is however mixed across principal components, as expected by the reported results in the previous subsection. This is due to the fact that while PCA analyses successfully capture the different sources of neuronal variability, this extraction is blind to the source of the variability. That is to say, this analysis does not allow to formally relate the extracted components to

specific task- or behavior-related parameters, and therefore these components are affected by mixed selectivity, in the sense that neuronal variability assigned to a specific neuronal process could be projected onto multiple PCA axes.

This issue is theoretically resolved by demixed principal component analysis (dPCA[28,29]), which retains the main objective of dimensionality reduction methods capturing almost as much variance as possible in only few latent variables or components, similarly to what PCA does, but without imposing orthogonality between components as PCA does by definition. Indeed, this property of the dPCA permits demixing the neural activity as a function of a priori-defined task parameters. As a result, this allows to interpret the internal dynamical and geometrical properties of neuronal population responses in terms of specific brain functions inferred from task design and behavior. In the following, we perform a dPCA analysis on the FEF MUA activity with the aim to extract unmixed components accounting for variability associated with the TA distance and the overt behavioral outcome, respectively. To do so, we pool the trials based on two different conditions: Upcoming target detection (hit vs miss) and attention (TA close vs. TA intermediate vs. TA far), giving rise to six different categories of trials. This analysis is performed on the epoch immediately preceding target presentation ([−300 0] ms with respect to target onset), thus focusing on the neuronal variability that best accounts for cue instruction processing and behavioral outcome.

Demixed PCA succeeded in reducing the dimensionality of the MUA data in components that held information associated with either TA (53% of explained variance), upcoming target detection (19%) and the interaction between them (23%). Little variance was associated with sources independent from these parameters or their interaction (5%) (Fig. 5A). The overall variance explained by the dPCA components is similar to the variance explained by the PCA components (Fig. 5B, left). Therefore, population activity is accurately represented by the dPCA components. Cumulative marginalized variance showed that the majority of the variance explained by each of the parameters and its interaction was accumulated in the first components (Fig. 5B, right). The projection of the MUA corresponding to each of the conditions onto each of the first dPCA components associated with TA and target detection captures similar findings as we observed in the MUA results (Fig. 5C, D): Just before target onset, projected firing rates onto the first demixed attention-related component dissociates each of the attentional states based on the TA, whereas the projection onto the component associated with target detection (i.e., behavioral outcome) shows two different states depending on whether the target was detected or not. In addition, we found an interaction component mostly associated with a neural population reflecting mixed selectivity between attention and target detection (Fig. 5E). These observations are globally reproduced on individual sessions (Supplementary Fig. 2).

To assess whether the tuning of each individual demixed component was statistically significant, we used these components as linear decoders to measure their ability to encode information associated with attention and behavioral outcome. We used cross-validation to measure time-dependent classification accuracy and a shuffling procedure to assess whether the accuracy was significantly above chance (see Materials and Methods). We found that the attention-related component achieved a single-trial classification accuracy of TA (TA close vs TA medium vs TA far, chance level: 33%) of 81%, whereas the behavioral outcome-related component achieved a classification accuracy of 62% averaged across time (chance level: 50%). All classification performances were assessed against the 95% confidence interval using a random permutation test. Interestingly, the interaction component achieved a classification accuracy of 30% (chance level: 16.6%).

Figure 5F shows that many neurons expressing one of these components tended also to express the other one, which is a marker of non-orthogonality of the demixed principal components (Spearman

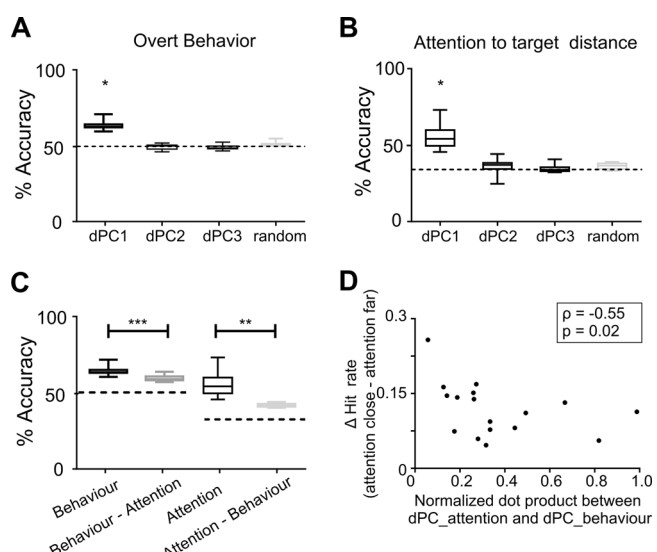

**Fig. 6 | Unmixed components account for both overt behavior- and attention-related information.** Box plot corresponding to the distribution of the cross-validated classification accuracies of linear classifiers given by the first three demixed principal components associated with upcoming behavioral outcome (**A**), and with attention to target distance (TA)(**B**). For each parameter, the horizontal dotted line corresponds to the theoretical chance level (50% for upcoming target behavior, and 33.3% for attention to target distance). Gray boxplot shows the distribution of classification accuracies expected by chance as estimated by 100 iterations of shuffling procedure (maximal accuracy value obtained across all iterations is considered). For both parameters, only the first dPC showed accuracy values above chance level (*$p < 0.05$, permutation test). **C** Box plot corresponding to the distribution of the cross-validated classification accuracies of linear classifiers given by the first demixed principal component associated with upcoming behavioral outcome and attention to target distance using two dPCA approaches: either demixing each parameter (upcoming behavioral outcome or attention to target distance) independently or demixing each parameter from the other (two-sided Wilcoxon signed-rank test, ***$p = 0.0008$; ** $p = 0.01$) (**D**). Scatter plot between the normalized dot product between the first demixed components associated with attention and upcoming behavioral outcome and the behavioral gain associated with an optimal attention orientation towards the upcoming target location (Spearman correlation, $\rho = -0.55$, $p = 0.02$). All panels of this figure contained independent measures corresponding of each session ($N = 18$ sessions), and lower and upper box boundaries in boxplots reflect the 90th and 10th percentiles, respectively, line inside the box reflects the median and lower and upper error lines show the min and max value of each distribution, respectively.

correlation, $\rho = 0.16$, $p = 8e-7$). To confirm this, and since the demixed components are not assumed to be orthogonal, we calculated the angle between the first encoding components associated with attention and behavioral outcome, respectively. The absolute value of the dot product between the attentional component and the target detection component is 0.22, which is higher than the significant non-orthogonality threshold of 0.123 (see ref. [28] for details). This confirms statistically the non-orthogonality of these two components. In addition, we observed that both components were equally distributed across the whole neuronal population, and the weights of each component showed a clear unimodal distribution centered close to zero (TA component, median = 0.03, IQR: 0.04; behavioral outcome component, median = 0.01, IQR: 0.047). This latter observation rules out the possibility that components might be exhibited only by a subset of cells.

All these results taken together point to the coexistence of two neural mechanisms that can be assigned to the reallocation of attention and to the upcoming behavioral outcome, respectively, and that can be reliably accessed at the signal-trial level. As expected by the dPCA theoretical framework, this method demixes the part of explained variance attributed to overt behavior and TA, respectively.

## Attentional performance depends on how attentional and behavioral outcome information are mixed in the prefrontal cortex

At the behavioral level, we observed an inter-session variability in how the TA modulated the hit rate of the monkeys (Supplementary Fig. 3). Indeed, in some sessions, attention close to the target position just before target onset induced a large increase in hit rates relative to when attention was far, while other sessions showed no clear benefit of attention orientation on the behavioral outcome. Therefore, we asked whether these differences in how TA affected behavior correlated with how attention and behavioral performance information interacted in the FEF. To address this question, we conducted dPCA in each session with the aim to find, at the session level, specific attention focus and behavior outcome components in the neural population. Similar to the previous section, we used the first dPC in behavioral outcome (hit or miss) and TA (close, intermediate, and far) obtained in each session as a decoding axis to build a linear classifier to decode these two variables. We found that the first dPCs associated with each of these parameters reliably encoded the expected information (Fig. 6A, B, behavioral outcome, median = 58.8%, min = 54.17%, max = 65.5%; attention to target distance, median = 54.2%, min = 45.5%, max = 73%). All classification performances were assessed against the 95% confidence interval using a random permutation test (Fig. 6A, B, light-dark box plot). By construction, in this dPCA, components the variance of which is highly related to one of the two parameters of interest contain minimal variance about the other one. Such a dPCA would not capture the neuronal variance that would be explained jointly by both parameters. To address this concern, we test whether 1) the information contained in the behavioral outcome component obtained using this dPCA (overt behavioral outcome and attention to target distance) is statistically undistinguishable from that estimated from a dPCA uniquely performed on behavioral outcome and 2) the information contained in the TA-related component using this dPCA is statistically undistinguishable from that estimated from a dPCA uniquely performed on TA.

The total cumulated variance explained by the six first dPC of the dPCA applied to the upcoming behavioral outcome parameter only was 97.7% (IQR: 1.9%). When dPCA was applied to TA parameter only, explained variance by the six first dPC was 96.1% (IQR: 4.3%). When decoding each of these parameters using the first dPC of each of these two dPCA (Fig. 6A, B), overall decoding accuracy across sessions was significantly higher than when using the first components of the two-dimensional dPCA that forced unmixed sources of variance (Fig. 6C, Behavior vs Behavior-Attention, two-sided Wilcoxon signed-rank test, $Z = 3.3$, $p = 0.0008$; Attention vs Attention–Behavior, Two-sided Wilcoxon signed-rank test, $Z = 2.57$, $p = 0.01$). This thus indicates that while part of the variance accounting for overt behavioral outcome and attention to target distance are independent, the remaining part is common to both parameters, thus resulting in a loss of decoding accuracy when the two sources of information are demixed. This observation indicates a functional overlap between the subspaces generated by the two components of interest. In order to quantify this functional overlap, for each session, we measured from the two-parameter initial dPCA, the normalized dot product between the first dPCs maximally related to overt behavioral outcome and to attend to target distance (as only these dPCs showed significant encoding information capacities for each category of interest, Fig. 6A, B). The median normalized dot product between the axis corresponding to the first dPC of each condition was 0.28 (IQR: 0.33). Most interestingly, the dot product between these two components negatively correlated with how much attention-to-target distance accounted for behavioral performance (Fig. 6D, Spearman correlation, $\rho = -0.55$, $p = 0.02$). In other words, the greater the functional overlap between the neuronal populations coding for attention to target distance and overt behavioral outcome, the lower the behavioral gain when attention is closer

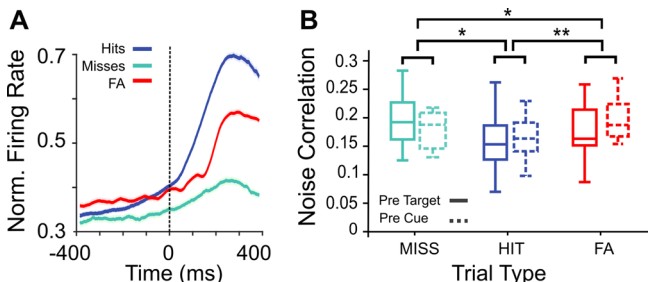

**Fig. 7 | FEF activity and noise correlation vary as a function of the type of produced behavioral responses. A** Average time evolution MUA ( ± s.e.m) across sessions recorded during hit (blue), miss (green), and false alarm (red) trials. Activity was locked to target (for hit and miss responses) or distractor onset (for false alarm responses) (0 ms, interval of analysis: −400 ms to 400 ms). **B** Boxplot representing the distribution across sessions of noise correlation values for each type of trial computed on pre-target (solid lines) or pre-cue (dashed lines) neuronal activities. Lower and upper box boundaries reflect the 90th and 10th percentiles, respectively, line inside box reflects the median, and lower and upper error lines show the min and max value of each distribution, respectively (*$p < 0.001$, **$p < 0.0001$, two-sided Wilcoxon signed-rank test).

to the target position as compared to far. This can be seen as an interference of overt behavioral outcome-related neurons with the actual coding of attention and suggests that optimal attentional performance would be observed for minimal functional overlap between the two neuronal populations.

## Prefrontal cortex encodes different levels of behavioral responsiveness

Although there is a very large literature on the contribution of the prefrontal cortex to spatial attention processes, there is, to our knowledge, no evidence of a specific neuronal process accounting for behavioral outcomes independently from, but possibly interacting with, attention orientation processes. In the following, we seek to identify such a neuronal process. To this aim, we categorize neuronal responses as a function of all different possible behavioral trial outcomes: hit trials (in response to target), miss trials (in response to target), and false alarms (to unexpected distractors). From a behavioral perspective, these trials can be ordered as no response trials (misses), controlled response trials (hits), and uncontrolled response trials (false alarms). These trials will be considered in this same order for the analysis of their distinctive neuronal correlates. The average firing rate of FEF neurons was differentially modulated as a function of trial type (Fig. 7A, see also[6]). In addition, and in agreement with previous studies[21,24], noise correlation prior to stimulus onset (target onset for hits and misses, and distractor onset for false alarms), computed over z-scored neuronal responses, varied as a function of behavioral outcome (Fig. 7B, solid boxes, Friedman test $X^2(18,2) = 19.11$, $p = 1e-6$). More specifically, hits accounted for the lowest noise correlation values, and noise correlation were higher in false alarms relative to hits and maximal in misses (Hits vs. Miss, two-sided Wilcoxon signed-rank test, $Z = 3.24$ $p = 0.001$; Hits vs. FA, Two-sided Wilcoxon signed-rank test, $Z = 3.4$ $p = 0.0007$; Miss vs. FA, two-sided Wilcoxon signed-rank test, $Z = 2.63$ $p = 0.008$). The statistical relationship between the noise correlation levels in each trial type was kept when performed on hit, miss, and false alarm trials that were equalized for TA metrics on the session level (Supplementary Fig. 4). Importantly, similar results were observed when noise correlation is calculated within the period prior to the presentation of the cue, prior to attentional deployment (Fig. 7B, dashed boxes, Friedman test $X^2(18,2) = 17.44$, $p = 0.00016$; Hits vs. Miss, Two-sided Wilcoxon signed-rank test, $Z = 3.5$ $p = 0.0005$; Hits vs. FA, Two-sided Wilcoxon signed-rank test, $Z = 2.8$ $p = 0.004$; Miss vs. FA, Two-sided Wilcoxon signed-rank test, $Z = 2.86$ $p = 0.003$). Given that these reported differences in noise correlation as a function of trial

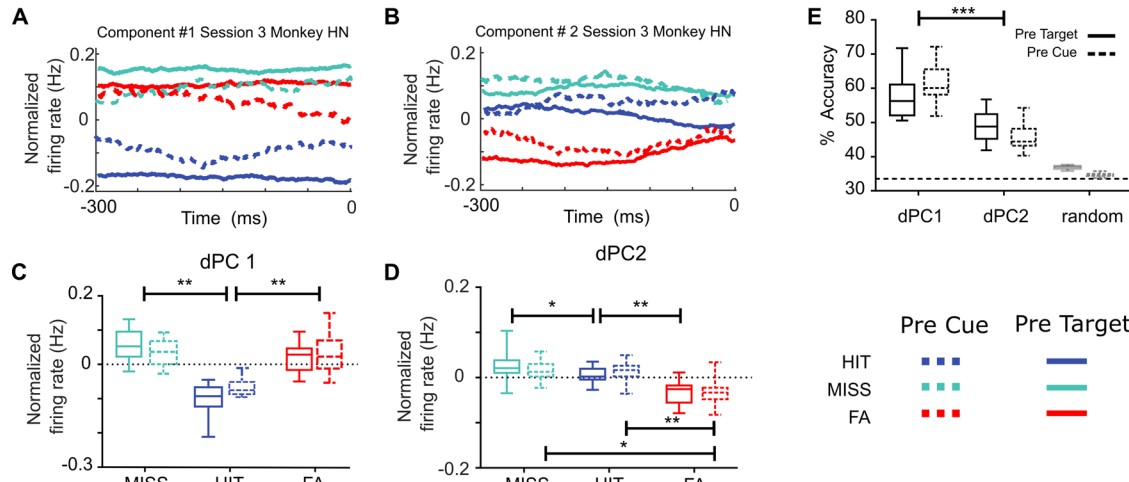

**Fig. 8 | Demixed PCA unmixes variance associated with trial types (hit, miss, and FA) in two independent components. A, B** MUA activity from the three different trial types (hit trials in blue, miss trials in green, and false alarm trials in red), computed on pre-target (solid lines) or pre-cue (dashed lines) neuronal activities, projected onto the first (**A**) and second (**B**) components that maximally explain trial type in one representative session, during the time period of −300 ms to 0 ms locked to the stimulus onset. **C, D** Boxplot representing the median and the interquartile range of the projected MUA activity onto the first (**C**) and the second (**D**) demixed components across sessions (two-sided Wilcoxon signed-rank test *p < 0.001; **p < 0.0001). **E** Box plot corresponding to the distribution of the cross-validated classification accuracies of linear classifiers given by the two demixed principal components associated with trial types shown in (**A**) and (**B**) (two-sided Wilcoxon signed-rank test, ***p = 8e-5). The horizontal dotted line corresponds to the theoretical chance level (33.3%). The Gray boxplot shows the distribution of classification accuracies expected by chance as estimated by 100 iterations of the shuffling procedure (maximal accuracy value obtained across all iterations is considered). Lower and upper box boundaries in boxplots reflect the 90th and 10th percentiles, respectively, the line inside the box reflects the median, and lower and upper error lines show the min and max value of each distribution, respectively.

type arise both during the cue to target interval as well as in the pre-cue period, we propose that they result from covariations in the excitability of the neurons, possibly due to more global variations in noradrenergic[38] or cholinergic[39–42] neuromodulation, rather than to covariations in response latency[43]. In addition, this result suggests the existence of different neural states that would predict the behavioral outcomes before stimulus onset and independent of ongoing attentional processing.

Our prediction is that these different trial types will differentially impact neuronal variability along a reduced number of behavioral outcome dimensions. To test this, we apply a dPC analysis to this neuronal data with the aim to find specific components in the neuronal population accounting for specific aspects of the variance associated with these different trial types. We focus on the epoch immediately preceding the target or distractor presentation ([−300 0] ms with respect to target or distractor onset), i.e., on the neuronal variability that best accounts for upcoming target/distractor processing. This dPCA thus respectively ranks the demixed principal components based on their explained variance either attributed to overt behavior (trial type) or to components independent from behavior. Figure 8A, B shows the projection of the high-dimensional MUA activity averaged over the three trial types (responses in hit, miss, and false alarm trials) onto each of the two first dPCs for one representative session. Across sessions, we found that these two dPCs associated with trial type accounted for two independent processes, and for 90.03% (IQR = 5.99) of the explained variance. The angle between these two components was 85.7 degrees (IQR = 6.33), and did not pass the test of non-orthogonality (see ref. 28). Importantly, both components were equally distributed across the whole neuronal population, and the weights of each component showed a clear unimodal distribution centred close to zero (dPC1 component, median = 0, IQR: 0.02; dPC2 component, median = 0.01, IQR: 0.031). This latter observation rules out the possibility that components might be exhibited only by a subset of cells (Supplementary Fig. 5). Projection of MUA activity onto the first component (Fig. 8C) showed that hit trials were in a different state as compared to both misses and false alarms (Friedman test, test

$X^2$ (18,2) = 27.1, p = 0; Hit vs Miss, Two-sided Wilcoxon signed-rank test, Z = 3.7, p = 0.0002, Hit vs FA; Hit vs FA, Two-sided Wilcoxon signed-rank test, Z = 3.7, p = 0.0002; Miss vs FA, Two-sided Wilcoxon ranked test, Z = 0.28, p = 0.7). Note that this projection mimics the results that we have obtained from the noise correlation analysis in these types of trials (Fig. 7B). The projection of MUA activity onto the second component (Fig. 8D) contrasts with the U-shaped curve identified in the first dPC, as MUA activity in each trial type organizes along a linear relationship (Friedman test, $X^2$ (18,2) = 27.1, p = 0.0006, Hit vs Miss, Two-sided Wilcoxon signed-rank test, Z = 2.4, p = 0.01; Hit vs FA, Two-sided Wilcoxon signed-rank test, Z = 3.2, p = 0.001; Miss vs FA, Two-sided Wilcoxon ranked test, Z = 2.9, p = 0.002). When using each of these two components as axes for decoding trial types, we find that performance in decoding trial types is above the 95% C.I. (Fig. 8E). Decoding performance is, however, significantly higher for the first component compared with that for the second component (Two-sided Wilcoxon signed-rank test, Z = 3.9 p = 8e-5).

A U-shaped relationship between behavior on the one hand and amplitude of the stimulus-evoked response or noise correlations, on the other hand, has already been described by others[44]. In order to better characterize the relationship between these two identified components (U-shaped dPC1 and linear dPC2, respectively) and noise correlation, we measured noise correlation levels in the three different trial types after removing the variance explained by one of the two components and projecting the pre-target activity contained in the rest of the components back onto the original data space. The differences in noise correlation between trial types vanishes when removing the variance associated with dPC1 but not when removing the variance associated with dPC2 (Supplementary Fig. 6a). This result indicates a functional link between noise correlation levels and the activity represented by the dPC1. Likewise, in order to better characterize the relationship between these two identified components (U-shaped dPC1 and linear dPC2, respectively) and stimulus-evoked response, we removed the variance associated with either the dPC2 (linear component) or the dPC1 (U-shaped component) estimated in the pre-target period, and we projected the stimulus-evoked response back onto the

original dataset space (Supplementary Fig. 6b). While firing rates extracted from miss trials and false alarm trials in post target time interval (corresponding to the stimulus-evoked responses) after removing the variance explained by the dPC1 did not differ relative to the original neuronal response, we observe that the firing rate differences during the time interval corresponding to the evoked response between miss trials and false alarm trials vanishes when we remove the variance explained by the dPC2. This is mostly accounted for by a change in the pre-target activity (Supplementary Fig. 6b). This indicates that none of the two dPCs account for the evoked response to the target but that dPC1 specifically accounts for pre-target differences in noise correlations and dPC2 specifically accounts for pre-target differences in average firing rate.

Crucially, all reported observations remained unchanged when the dPCA was applied on pre-cue activities, prior to task-related attention orientation (Fig. 8, dashed symbols, Supplementary Fig. 6). The only observed difference is a less marked linear trend in dPC2 between miss, hit, and FA trials in pre-cue as compared to pre-target analysis, although this difference disappears when the dPCA is performed on cumulated session data (Supplementary Fig. 7) rather than on single-session data (Fig. 8). In addition, we tested whether the firing rates projected in these two dPC (per each trial type) were linked to the activity obtained in the previous trial (irrespective of the trial type of the preceding trial). In this context, we found that the firing rate projected in the behavior-related components correlated with the activity of the preceding trial projected in the same space (Supplementary Fig. 8). These results indicate that the demixed components identified based on behavioral outcome can be identified irrespective of specific task-related processes and describe a neuronal process that extends across trials. This nicely dovetails with previous findings suggesting a low-frequency dynamics of the activity of a state linked to sensory processing efficiency, encompassing multiple trials over temporal scales of several minutes[12,45]. Likewise, none of the results reported in this section changed when performed on hit, miss and false alarm trials that were equalized for TA metrics on the session level (Supplementary Fig. 4). Overall, this thus indicates that, independently from cued position or attention orientation in the trial (TA), the different trial types are associated with a distinct structure in neuronal variability that can be tracked before target onset, but also prior to cue presentation. This strongly suggests that these specific neuronal states can be associated with distinct behavioral states that are predictive of behavioral outcome.

## Discussion

In the present work, we show that the accurate performance in a visual attentional task does not exclusively depend on attentional orientation signals, but also on the integration of these signals with pre-existing activity associated with neural states that modulate different levels of distractibility (defined as an absence of responses) and impulsivity (defined as inappropriate responses), optimality and responsiveness, that directly affect how attentional processes are implemented. We show that these two distinct behavioral markers, i.e., optimality and responsiveness, are implemented in two functionally different neuronal population components, the variance of which is partially dependent with the variance associated with the decoded position of the attentional spotlight relative to the target. These components are not orthogonal, indicating an overlap in how neuronal populations implement the information from each of these two components. Importantly, smaller overlap was associated with enhanced behavioral gain of efficient attention orientation (Fig. 9A). Furthermore, we find that this behavior-related neural state is two-dimensional, indicating that activity in the FEF associated to this neural state possibly reflect the effect of the activation of two independent input sources (Fig. 9B). Overall, these results indicate that the behavioral outcome during a covert attentional task can be attributed to multiple neuronal

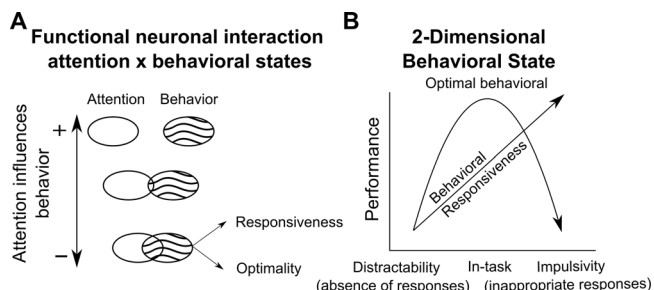

**Fig. 9 | Schema of the neurophysiological underpinnings explaining the relationship between covert attention and overt behavior. A** Behavioral gain produced by the good allocation of attention with respect to expected target position varies as a function of the level of overlap between the functional population associated to covert attention (blank ellipses) and overt behavior (textured ellipses). **B** This latter component is associated with two pre-existing neuronal states describing two task-independent behavioral states, reflecting the degree of distractibility (no response)-to-impulsivity (inappropriate response) or responsiveness of the subject (linear) as well as degree of behavioral optimality in the task (U-shaped).

processes in addition to spatial orientation processes and is a consequence of the interaction between task-specific computations with global neural states associated with different levels of impulsivity and distractibility that, in turn, influence the access to attention information.

### FEF contains mixed-selectivity cells that simultaneously encode overt and covert behavior

At the neuron level, we show that both individual neurons and population activity have higher spiking rates prior to target onset in upcoming hit trials than observed during the same period in upcoming misses. These results are in line with prior studies showing slight differences between the firing rates of the FEF cells based on whether the target was reported or not[16,18,19]. While these differences are often taken as a signature of spatial attention orientation, we further report that pooling trials based on attention to target distance, spiking rate of selective cells increased as closer was the attentional spotlight to the expected position of the target, in agreement with previous studies indicating that FEF plays a key role in attentional control[1]. These results indicate that FEF neural population encodes both upcoming target behavior and attention to target distance. FEF cells show different coding strength for each parameter, some preferentially encoding one of these two parameters (25.2% attention cells vs 9.2% upcoming perception cells) and some encoding both parameters simultaneously (~14%; mixed selectivity cells). These mixed selectivity cells have been reported in different brain areas such as the prefrontal cortex[36,37], parietal cortex[46] and visual cortex[47], and they represent a signature of high-dimensional neuronal representation of relevant information[37]. An important question in this context is whether the neuronal dimension encoding upcoming behavior interferes with attentional coding or whether it organizes independently (orthogonally). This is discussed next.

### Functional subpopulations associated with attention and upcoming behavior overlap

In the last decade, dimensionality reduction techniques have been applied on high-dimensional neural recordings involving a big number of recorded neurons, in order to provide a low-dimensional data-representation containing the functional structure of the data[48]. In particular, this approach allows to describe how independent cognitive functions are implemented in any given neuronal population, whether they are expected to interact or not and, if applicable, whether this interaction interferes with behavior or not. Due to the mixed-

selectivity observed at the neuronal level, dimensionality reduction of our data using PCA, resulted in partially mixed selectivity of the three principal components with respect to our parameters of interest. In order to find a low-dimensional decomposition of the population data, that is interpretable in terms of our variables of interest, we applied demixed-principal component analysis to our data[28,29]. With this method, we found two specific components that were associated with either upcoming behavior or attention to target distance. Importantly, we found that these two components could be used as a decoding axis for target perception and attention, respectively, indicating that the reported source of the variance was functionally meaningful. Two important findings were achieved by the application of this method. First, we found a component specifically associated with the interaction between covert attention and overt behavioral response. This component successfully decoded both parameters simultaneously, which is a signature of non-linear mixed selectivity[37]. Non-linear mixed selectivity is advantageous as it allows the information to be explicitly distributed across multiple neurons. In the present context, this means that information related to how close the attentional spotlight is to the target is encoded differently as a function of the upcoming target behavior prior to the target onset. This thus indicates the existence of a neuronal state that impacts in the way the FEF encodes attention information. A second finding relies on the non-orthogonality of the linear readouts of behavioral outcome and covert attention related components. In order to associate each component to a specific known source of variance, demixed-PCA relaxes the assumption of orthogonality between components[29]. Therefore, components that contain variance attributed to different parameters might be non-orthogonal. This is exactly what we found in our data. Our analysis revealed that components encoding attention and upcoming behavior were statistically non-orthogonal, which predicts that encoded information for each parameters is, at least partially, overlapped. In agreement with our prediction, we found that subtracting the variance attributed to one parameter impacted the decoding performance of the other parameter negatively. We predicted that such overlap would impact how the two parameters functionally interact. Accordingly, we found a correlation between the degree of functional overlap between these two parameters and the behavioral gain (reported by the hit rate variation) resulting from attention being closer to compared to distant from the target. In other words, the higher was the overlap at the neuronal level, the higher was the interference at behavioral level between the orientation of the spatial attention and the behavior-related state encoded by the prefrontal neuronal population (Fig. 9A). This result indicates that responding to or missing the target might be not only a consequence of the localization of the attentional spotlight with respect to the stimulus, but also a consequence of how such behaviorally-related neural state affects the implementation of attentional control. In this context, a precise behavioral definition of this behavioral outcome state-related component becomes paramount.

## FEF encodes two independent neuronal states associated with behavioral outcome in the trial

Prior studies have shown that task-independent neural states might influence different aspects on how information is encoded in a neural population and, hence, its impact in behavior[11,12,21]. For example, the level of shared noise correlation between neurons predicts subjects' behavior[21,25,49]. In particular, Astrand and colleagues[21] show that high levels of noise correlation predict misses and false alarm responses, while hit responses are produced when noise correlation level is low. Importantly, these levels of noise correlation predict behavioral outcome irrespective of spatial orientation processes. This is in line with yet other studies which have identified different states in the population activity associated with differences in evoked responses and correlated variability[44]. This suggests that these levels of correlation possibly describe a functional state of the prefrontal neuronal

population. The present work reproduces these observations on an independent dataset and extends these results. Using demixed principal component analysis on data locked to the stimulus onset (locked to the target for hit and miss responses, and locked to the distractor onset for false alarm responses) we found a two-dimensional representation of the population in the FEF that encoded upcoming behavioral responses to task-relevant (target) and task-irrelevant (distractor) stimuli. Specifically, we found two demixed components that contained significant information about upcoming behavior at trial bases. Importantly, these two components were close-to-orthogonal (~86 degrees), indicating that these components share no significant variance. The projection of the firing rates related to each behavioral condition onto each of these two different demixed behavior-related components showed different activations across the different types of trials. When projected onto the first component, normalized firing rates of each condition showed a U-shape, in which lower activity state corresponded to hit trials, whereas the activity in miss and false alarm trials showed higher activity states. In contrast, projection of the activity from each of the three behavioral conditions onto the second demixed component revealed a linear decrease of the activity state, being higher in miss trials, intermediate in hit trials and minimal in false alarm trials. Overall, this points to two independent neuronal processes contributing to behavioral outcome in addition to the classically-described attentional orientation (Fig. 9B). None of the two components account for the evoked response to the target. However, the U-shaped component specifically accounts for pre-target differences in noise correlations and the linear component specifically accounts for pre-target differences in average firing rate. Importantly, these two processes did not depend on the implementation of spatial attention following cue presentation, but rather expressed themselves at a longer time scale. Indeed, they could be reliably identified both before cue onset and during the cue to target interval. We propose that these two neuronal functional states map onto two distinct behavioral states that impact overall performance in the task. This is discussed next.

## U-shaped state of inattention, in-task behavior, and distractibility

The U-shape observed in the projection onto the first behavioral outcome component of FEF neuronal population firing rates accounts for the U-shape observed for noise-correlation levels associated with the same behavioral conditions (Fig. 9B, Supplementary Fig. 6a). This indicates that the dPCA captures specific variance explained by the different levels of shared variability between neurons within the same population. How is this U-shape functional component generated? We argue that these different states represented by each trial type correspond to different patterns of activity of the noradrenergic neurons in the locus coeruleus (LC-NE), a subcortical structure that is described to play a critical role in adaptive behavioral decision-making[50] and arousal (see below). Indeed, prior studies have shown that these LC neurons are projected onto the medial and posterior prefrontal cortex via well-identified anatomical pathways and are thus expected to modulate neuronal activity in these regions[51]. In this context, Aston-Jones and Cohen[30] propose that two different modes of LC activity might correspond to different patterns of performance in attention tasks. A first phasic mode is described, in which LC-NE cells show a phasic activity in response to task-relevant stimuli. Within this mode, LC-NE cells show a moderate level of tonic discharge associated with high levels of task performance. A second tonic mode is described, which has been associated with poor performance, whether characterized by high levels of distractibility or by high levels of inattention. In addition, prior studies in mice have found that molecular down-regulation of phosphoinositide 3-kinases enzymes in LC-NE cells results in both increased levels of tonic activity and increased levels of inattention[52], compatible with symptoms associated with attention deficit hyperactivity disorder

(ADHD). Thus, LC-NE activity organizes along U-shaped curve similarly to what we described in the FEF first behavioral outcome component, such that its tonic baseline activity is high during inattention (misses), low during optimal behavioral (hits, in-task) and high again during distraction (false alarms). We would thus like to propose that this component is associated with the level of optimal (in-task) behavioral regime as described in the Locus Coeruleus noradrenergic neurons[30].

Prior studies have argued that the different levels of the NE system might mediate arousal states[53,54]. However, other studies refine the understanding of the LC-NE function by relating it with optimization of reward-seeking behavior[30]. Indeed, although the term arousal is broadly used to describe a mental state associated with different levels of consciousness and alertness, it has proven difficult to link it to specific neurobiological mechanisms[30]. Therefore, we would like to remain cautious when trying to associate our findings with different levels of arousal, in the absence of more direct causal evidence (see below). Indeed, the causality of the relationship between the FEF U-shaped functional component that we report in our data and LC-NE tonic discharge rate that we suggest here is very indirect at this stage and will have to be tested experimentally.

## Linear state of executive control and responsiveness
The second demixed component of the dPCA performed on behavioral outcome showed a negative linear trend between the level of activation state and behavioral responsiveness, defined as the probability of the subject issuing a behavioral response to a stimulus. Accordingly, miss trials correspond to low responsiveness trials, hit trials to optimal responsiveness trials and false alarm trials to high responsiveness trials. Along this component, higher levels of activity were observed in miss trials, followed by intermediate levels of activity in hits and low activation in false-alarms. The neural state defined by this second dPCA component is thus inversely associated with the level of impulsivity in the response. Our current data cannot allow us to provide clear-cut responses on the origin of this state. It could arise locally in the FEF. Alternatively, it could originate from outside the FEF. Overall, we propose that this component is associated with the level of executive control (or responsiveness) exerted by the subject in the task (Fig. 9B). Recent studies suggest that the level of impulsivity in motor responses during attentional tasks is associated with the activity of the claustrum, a brain region located deep to the insular cortex and extreme capsule with anatomical projections toward multiple cortical brain areas including motor, visual, auditory and prefrontal cortices[55,56]. In particular, the activity of this area has been shown to play a critical role in attention. Goll and colleagues[55] posit that the claustrum might act to control the output from the cortical representation of the sensory modalities. It has been shown that electrical activation of the claustrum causes irresponsiveness[57], while its deactivation prevents tuning down the output from cortical modalities irrelevant to the ongoing task[55]. Other studies have shown that the claustrum-prefrontal cortex pathway regulates methamphetamine-induced impulsivity, suggesting a critical role of this pathway in regulated impulsivity-related disorders[58]. Our results are compatible with the hypothesis that the level of responsiveness state observed in our data reflects the level of motor impulsivity in attentional tasks and is driven by the claustrum. This will need to be addressed in future studies.

## Physiological correlates of the behavior-related components
In the previous section, we have argued that the different levels of activation of the neural states associated with optimality and responsiveness contribute to the processing of sensory information and its behavioral consequences. However, there is a lack of consensus on which physiological mechanisms might correlate with such state changes and account for either an optimal or sub-optimal sensory encoding or behavioral performance. Prior studies have shown that pupil size might be a potential candidate as a proxy of these fluctuations in the state, as it has been shown that it impacts both the sensory evoked response, the spontaneous activity of cortical responses[41], as well as on membrane potential level[59]. Indeed, McGinley and colleagues[59] have shown, in mice, that optimal signal detection behavior and evoked responses occur at intermediate arousal (measures by pupillometry) when pre-decision membrane potentials are hyperpolarized, revealing a cortical physiological signature of the inverted-U relationship between performance and arousal. Relating to these findings, we have recently described consistent low-frequency fluctuations in the pupil size from human data recorded during the execution of an exploration task, in the order of a few cycles per hour. These rhythmic fluctuations are linked with different behavioral states characterized by differences in detection times as well as in prefrontal attention and perception information capacity[12] and, possibly, the activity of the noradrenergic system[60].

Another possible physiological correlate of the activity reflecting these neural states could be the global network influence on the local functional connectivity. In this context, Snyder and colleagues[61] showed that spike count correlation in area V4 of the macaque correlated with certain properties of the EEG, and specifically with the level of alpha-band oscillatory activity measured in different regions of the occipital cortex, as well as with the reaction times during a spatial attention task. Relevant to the present discussion, this relationship between the EEG alpha oscillation and the spike count correlation and behavior were U-shaped. Other studies have shown the same inverted U-shape relationship between EEG oscillations during attention-demanding detection tasks and performance[62–64].

All in all, our results are in line with previous studies showing a non-monotonic relationship between different levels of activity linked with brain state, measured with different physiological measurements and specific behavioral regimes.

## Implications in attention deficit and hyperactivity disorder (ADHD)
The present work provides consistent evidence on how different neural states associated with levels of distractibility and impulsivity interact with dynamic, ongoing computations in the attentional system to produce behavior. We believe that these observations have a profound implication in the way we understand how the attentional system works and, more particularly, how it can dysfunction. One clinical condition affecting the attentional system is attention deficit and hyperactivity disorder (ADHD), with a prevalence of 7.5% of the worldwide population[65]. This disorder is specifically characterized by a dysfunction of attention as well as by inappropriate levels of hyperactivity, distractibility and impulsivity[66]. Many studies point that ADHD patients show abnormal results in neuropsychological tasks targeted to measure sustained and focused attention[67,68], however other studies do not find significant differences in tasks requiring orienting of attention compared with healthy population[15]. This would indicate that selective visual attention remains functionally intact in these patients and, in line with our results, their behavioral symptoms might arise from the dysfunction of non-attentional neural populations, interfering with the subject's access to attentional information. In addition, prior studies have found that ADHD patients show hyperactivation of the LC-NE system[69], as well as higher activation in claustrum[70], which correspond to a compromised state of alertness and increased processing of task-irrelevant information, respectively. This thus hints towards an abnormal interaction between attention orientation processes and other independent processes associated with aberrant distractibility and impulsivity behaviors observed in these patients. However, more studies are needed to confirm the implications of our findings in such a clinical domain.

In conclusion, we report converging evidence indicating that the access to attention-related processes in the FEF is driven by the activity

of a two-dimensional neural state that might be explained by the independent activation of afferent brain regions modulating the levels of distractibility and impulsivity as well as the levels of behavioral responsiveness. This finding sheds light onto the understanding of the computational mechanisms of the attentional system, and it is expected to have profound implications in the development of rehabilitation strategies to ameliorate inattention and impulsivity symptoms in ADHD patients.

## Methods

### Subjects and surgical procedures

Two adult male rhesus monkeys (Macaca mulatta), weighing 8 kg (monkey D, 6 years old) and 7 kg (monkey HN, 7 years old), contributed to this experiment. Both monkeys underwent a unique surgery during which two MRI-compatible recording chambers were implanted over the left and the right FEF hemispheres, respectively, as well as a head fixation post. A 0.6 mm isomorphic anatomical MRI scan was acquired post-surgically on a 1.5 T Siemens Sonata MRI scanner, while a high-contrast oil-filled 1mmx1mm grid was placed in each recording chamber in the same orientation as the final recording grid. This allowed precise localization of the arcuate sulcus and surrounding gray matter underneath the recording chambers. The FEF was defined as the anterior bank of the arcuate sulcus, and we specifically targeted those sites in which a significant visual and/or oculomotor activity was observed during a memory-guided saccade task at 10° to 15° of eccentricity from the fixation point. All surgical and experimental procedures were approved by the local animal care committee (C2EA42-13-02-0401-01) approved by the French Ministry of Research and in compliance with the European Community Council, Directive 2010/63/UE on Animal Care.

### Endogenous cued detection task and experimental setup

The task is a 100% validity endogenous cued luminance change detection task (Fig. 1A). The animals were placed in front of a PC monitor (1920 × 1200 pixels, refresh rate of 60 Hz) with their heads fixed. Stimulus presentation and behavioral responses were controlled using Presentation® (Neurobehavioral Systems, Inc.). To start a trial, the monkeys had to hold a bar placed in front of their chair, thus interrupting an infrared beam. The appearance of a central fixation cross (size 0.7° × 0.7°) at the center of the screen, instructed the monkeys to maintain their eye position (Eye tracker - ISCAN, Inc.) inside a 2° × 2° window, throughout the duration of the trial, so as to avoid aborts. Four gray landmarks (LMs size 0.5°×0.5°) were displayed, simultaneously with the fixation cross, at the four corners of a hypothetical square having a diagonal length of ~28° and a center coinciding with the fixation cross. The four LMs (up-right, up-left, down-left, down-right) were thus placed at the same distance from the center of the screen having an eccentricity of ~14°. After a variable delay from fixation onset, ranging between 700 to 1200 ms, a 350 ms spatial cue (small green square - size 0.2° × 0.2°) was presented next to the fixation cross (at 0.3°), indicating the LM in which the rewarding target change in luminosity would take place. Thus, the cue presentation instructed the monkeys to orient their attention towards the target in order to monitor it for a change in luminosity. The change in target luminosity occurred unpredictably between 750 to 3300 ms from cue onset. In order to receive their reward (a drop of juice), the monkeys were required to release the bar between 150 and 750 ms after target onset (hit). To test the monkeys' ability at distractor filtering, on half of the trials, one of the two distractor typologies was randomly presented during the cue-to-target delay. In ~17% of the trials (D trials), a change in luminosity, identical to the awaited target luminosity change, took place at one of the three uncued LMs. In these trials, the distractor D was thus identical in all respects to the expected target, except for being displayed in an uncued position. In ~33% trials (d trials), a local

change in luminosity (square) was displayed at a random position in the workspace. The size of the local change in luminosity was adjusted so as to account for the cortical magnification factor, growing from the center to the periphery. In other words, d distractors had the same size as D distractors when presented at the same eccentricity as D. The absolute luminosity change with respect to the background was the same for both d and D. The monkeys had to ignore both distractor typologies (correct rejections – RJ). Responding to such distractors within 150 to 750 ms (false alarm - FA) or at any other irrelevant time in the task interrupted the trial. Failing to respond to the target (miss) similarly aborted the ongoing trial.

### Electrophysiological recordings and spike detection

Bilateral simultaneous recordings in the FEF in both hemispheres were carried out using two 24-contact Plexon U-probes (Fig. 1B). The contacts had an interspacing distance of 250 µm. Neural data was acquired using a Plexon Omniplex® neuronal data acquisition system. The data was amplified 500 times and digitized at 40,000 Hz. Neuronal activity was high-pass filtered at 300 Hz and a threshold defining the multiunit activity (MUA) was applied independently for each recording contact and before the actual task-related recordings started.

### Decoding procedure

**Training procedure.** In prior studies, we showed that the endogenous orienting of attention (Fig. 1C) can be reliably decoded from the FEFs activity using a regularized optimal linear estimator (RegOLE) with the same accuracy as exogenous visual information[21,23,71–73]. Here, we used the same approach to train a RegOLE to associate the neural responses prior to target onset ([−220 + 30] from target onset), based on a leave-one-out training/testing procedure, with the attended location, i.e., with the expected target presentation LM, based on cue information. Neural responses consisted in a vector containing the MUA signals collected at each of the 48 recording contacts during this pre-defined pre-target onset epoch. Our general objective here was to have as precise as possible an estimate of the attention position before a specific visual event, averaging activities over large enough windows to have a reliable single-trial estimate of the neuronal response on this window, while at the same time a not-too-large time window to have a reliable estimate of where attention was placed by the subject at a specific time in the task[22,23,73].

The RegOLE defines the weight matrix W that minimizes the mean squared error of **C**=**W**\*(**R**+**b**), where **C** is the class (here, four possible spatial locations), **b** is the bias, and **R** is the neural response. To avoid over-fitting, we used a Tikhonov regularization[71] which gives us the following minimization equation $\|\mathbf{W}*(\mathbf{R}+\mathbf{b}) - \boldsymbol{C}\|^2 + \lambda*\|\mathbf{W}\|^2$ .

The scaling factor $\lambda$ was chosen to allow for a good compromise between learning and generalization. Specifically, the decoder was constructed using two independent regularized linear regressions, one classifying the x-axis (two possible classes: −1 or 1) and one classifying the y-axis (two possible classes: −1 or 1).

**Testing procedure.** In order to identify the locus of attention at the moment of target or distractor presentation in the 20 next new trials following the initial training set, the weight matrix defined during training were applied to the average neuronal activity recorded in the 150 ms prior to the target. The described training (over 200 previous trials) / testing (over 20 novel trials) procedure was repeated after every 20 correct responses by re-training the decoder with the new database composed of the last 200 correct trials. This continuous updating of the weight matrix W is implemented in order to minimize the impact of possible uncontrolled changes in the recorded signal during a given recording session onto the decoding procedure.

## Estimating the (x,y) spatial locus of the attentional spotlight (AS)

As in Astrand et al.[21], the readout of the RegOLE was not assigned to one of the four possible quadrants by applying a hardlim rule, as usually done for classification purposes. Rather, it was taken as reflecting the error of the decoder estimate to the target location, i.e., in behavioral terms, as the actual (x,y) spatial estimate of the locus of the attentional focus to the expected target location. We show here and elsewhere[21,23] that this (x,y) estimate of the attentional spotlight (AS) accounts for variations in behavioral responses. In order to analyze how the distance of the decoded attentional spotlight to the target affected both behavior and neuronal MUA responses, we computed, for each target presentation, the distance between the decoded AS and the target (TA) as follows: $TA = \sqrt[2]{(x_{AS} - x_t)^2 + (y_{AS} - y_t)^2}$ where $x_{AS}$ and $y_{AS}$ correspond to the Cartesian coordinates of the attentional spotlight (AS), and $x_t$ and $y_t$ correspond to the Cartesian coordinates of the target position (T).

## Characterizing MUA selectivity

In order to quantify the magnitude of the modulation of FEF individual neurons to different task variables, we computed three different indexes per neuron, as follows: 1) RF-based attention index, pooling trials based on whether the cue oriented attention in the preferred spatial location within the neuron's receptive field (RF), or non-preferred spatial location outside the RF, considering only correct trials, 2) Hit/miss modulation index: pooling trials based on whether the monkey produced hits or misses, irrespective of the TA distance or cue position, 3) Attentional spotlight modulation index, pooling trials based on whether the target to focus of the attention distance was smaller than 6° or larger than 12°, considering only correct trials. For this latter measurement, we binned the trials in three different categories: TA close (0° < TA ≤ 6°), TA medium (6° < TA ≤ 12°) and TA far (12° < TA < 18°). For each of these trial categories, the modulation index was defined as $MI = \frac{(FR_{class1} - FR_{class2})}{(FR_{class1} + FR_{class2})}$ where $FR_{class1}$ and $FR_{class2}$ correspond to the median firing rate of each for each of the two classes defining the index. Firing rates were computed on the [−250 − −50] ms pre-target epoch, z-scored with respect to a [−100 − 0] ms pre-cue epoch. For each category, significant difference between the neuronal firing rate in each class was assessed using a Wilcoxon non-parametric test. In addition, for each of these trial categories, data were averaged from −400 ms to 400 ms locked to the target onset and median MUA activity as well as standard error (s.e.) was computed across all MUA channels and all sessions.

## Noise correlation measurements

In order to quantify the spiking statistics of the FEF activity associated with different overt behavioral outcomes (hit, miss, false alarm, correct rejection to distractors in hit, and correct rejections to distractors in miss trials), we measured the noise correlation between the MUA activities on the different simultaneously-recorded signals. For each session and for each channel, we defined intervals of interest of 200 ms previous to the stimulus onset (target or distractor). For each channel $i$, and each trial $k$, the average neuronal response $r_i(k)$ for this time interval was calculated and z-scored within this time interval. Noise correlations between pairs of MUA signals during the interval of interest were defined as the Pearson correlation coefficient between the z-scored individual trial neuronal responses of each MUA signal over all trials. Only positive significant noise correlations are considered.

## Demixed PCA

Recent research points that neural function is built on population activity patterns rather than on independent modulation of individual neurons[48]. These patterns reflect the coordination of responses across neurons that corresponds to a specific neural mechanism underlying behavior[35]. The population activity structure can be estimated by applying a dimensionality reduction technique to the recorded activity, such as principal component analysis (PCA). Using this method, we can extract a number of latent variables (principal components) that capture independent sources of data variance, providing a description of the statistical features of interest[48]. However, this method is blind to the source of variability of the data and hence does not take task- or behavior-related parameters into account, mixing these sources of information within each of the extracted latent variables[28,29].

Our goal here is to describe how much variance in the neural population can be explained by spatial attention and the overt behavioral outcome and their interaction. To do so, we performed a demixed principal component analysis[28] which captures the maximum amount of variance specifically explained by defined sources of variability in each extracted latent variable (or component) and reconstructs the time course of the category-specific response modulation. In a first dPCA analysis, trials were thus segregated as a function of classes of TA distance (TA close, TA medium, TA far) and, within these classes, trials were pooled into two possible trial outcome classes (hit or miss). This thus resulted into a six different conditions. In a different analysis, we extracted attention and overt behavior-related components by unmixing each of the two categories from category-independent variability in two distinct dPCA analyses (one targeting attention, and the other on overt behavior). In a last dPCA analysis, trials were segregated in three possible classes of trial (hit, miss, false alarms).

Procedures for dPCA analysis were performed using the MATLAB© (The Mathworks Inc., Natick, Massachusetts) written scripts available from[28]. Spike trains were filtered with a Gaussian kernel ($\delta$ = 30 ms) and averaged over all trials to obtain smoothed average pre-stimulus (target or distractor, 400 ms before target onset to 0 ms) MUA firing rate for each channel in each condition and each session. In this case, dPCA decomposes data into latent variables that estimate independently over time both the variance attributable to the specific categories of interest (attention or overt behavior) as well as the variance independent of any of the considered category. We consider the 20 demixed components that accounted for > 90% of the total variance in all sessions. Last, we used the decoding axis of each dPC assigned to each category (attention or overt behavior) as a linear classifier to decode the different types of trial (details of this procedure are fully described in[28]. This method allows the understanding of the capacity of each demixed component to classify a trial between the classes of a given category. To extract the statistical significance of this accuracy, we shuffled 100 times all available trials between classes and we thereby computed the distribution of classification accuracies expected by chance. For each session, the chance-level was considered as the maximal accuracy value obtained across all randomizations.

The different components extracted by the dPCA are not assumed to be orthogonal, therefore, we calculated the dot product between each encoding axis related to attention or overt behavior that showed significant levels of decoding accuracy for the corresponding category. Because the coordinates of the components reflect the level of contribution to the activity of each neuron, the size of the dot product values between two components indicates that neurons that contribute to one component tend also to contribute in the other component. Therefore, the dot product between two components can be interpreted as a marker of functional overlapping between the two different components. For each session, we calculated the dot product between the pairs of encoding (showing above chance-level accuracy) demixed principal components relative to attention and overt behavior. We considered the absolute value of the dot product.

## Reporting summary

Further information on research design is available in the Nature Research Reporting Summary linked to this article.

## Data availability
Source data are provided with this paper.

## Code availability
Codes supporting these results are available upon reasonable request to the corresponding author.

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

## Acknowledgements

J.A., F.D.B., and S.B.H. were supported by ERC BRAIN3.0 # 681978 and ANR-11-BSV4-0011 & ANR-14-ASTR-0011-01, LABEX CORTEX funding (ANR-11-LABX-0042) from the Université de Lyon, within the program Investissements d'Avenir (ANR-11-IDEX-0007) operated by the French National Research Agency (ANR) to S.B.H. We acknowledge the fruitful scientific discussion with Dr. Fardin Afdideh and Dr. Corentin Gaillard on this manuscript. We acknowledge Inovarion for their fruitful scientific discussions. We acknowledge the comments of the two independent reviewers, which have helped to increase the quality of our work. We also extend our thanks to Serge Pinède for hardware and software assistance and Jean-Luc Charieau and Fidji Francioly for animal care.

## Author contributions

Conceptualization, J.A. and S.B.H.; Data Acquisition, F.D.B. and S.B.H.H.; Methodology, J.A., F.D.B., and S.B.H.; Investigation, J.A., F.D.B., and S.B.H.; Writing – Original Draft, J.A. and F.D.B.; Writing – Review and Editing, J.A.

## Competing interests

The authors declare no competing interests.
