## [Peer Review File · Nature Communications]

Distractibility and impulsivity neural states are distinct from selective attention and modulate the implementation of spatial attentionREVIEWER COMMENTS

Reviewer #1 (Remarks to the Author):

The authors have studied the neural basis of spatial attention in prefrontal neurons.

The authors show that in addition to encoding spatial attention, prefrontal neurons also encode a distractibility-to-impulsivity state. The authors postulate that the pre-existing neuronal states describing task-independent behavioral states sheds new light on attention disorders such as attention deficit-hyperactivity disorder.

The experiments are straightforward, the manuscript is written clearly and should clearly be consider for publication in this journal. The conclusions are reasonable. However, I have some questions regarding the methodological treatment of the data.

Major comment:

1- The authors focused their analysis on the epoch immediately preceding the target or distractor presentation ([-300 0] ms with respect to target or distractor onset). Therefore, trial history and the role of the preceding trial could be critical. Would it be possible to relate their finding to possible activity in trial t-1?

2- In his paper Brody have shown that two biologically plausible ways of departing from independence can generate peaks very similar to spike timing peaks are described: covariations over trials in response latency and covariations over trials in neuronal excitability. Could the authors state over which plausible causation they observed some noise correlation in FEF.

Brody CD. Correlations without synchrony. *Neural Comput.* 1999 Oct 1;11(7):1537-51. doi: 10.1162/089976699300016133. PMID: 10490937.

Reviewer #2 (Remarks to the Author):

The manuscript builds on top of an impressive body of literature from the same lab, using decoding methods of neuronal activity recorded in the frontal eye fields of macaque monkeys to get an estimate of the attentional focus of the subjects. Observing that this estimate can not explain all variability in the data, the authors use the behavioral response made by the subject (hits, misses and false alarms) to decode the brain state of the subjects. They use a demixed principal component analysis (dPCA) to investigate these two components (attention and behavioral response) in more detail and find that they are partly overlapping and partly orthogonal. Interestingly, the distance of the attentional focus relative to the target location ('Target-Attention distance', or TA) is negatively correlated with the overlap between the two components. In a separate dPCA they analyze the behavioral components in more detail and report two main components (dPC1 and dPC2), one that seems to reflect (the inverse of) arousal and the other that corresponds to the effect of arousal on performance, a U-shape with an optimum at mid-levels of arousal (similar to the Yerkes-Dodson curve).

This is a highly original work, using state-of-the-art recording techniques and analysis methods to disentangle the different neuronal population dynamics that determine performance on an attentional task. The methods are generally sound, and the figures are clear. I therefore strongly support publication of it.

At the same time, some aspects of the paper can be improved, in particular the readability and the relationship to previous literature on the topic.

General comments:

1) The text is sometimes a bit convoluted, and the terms used to interpret the results are not always clear. Related to that, reference to some of what seems to me like the most relevant literature is lacking, in particular the classic Yerkes-Dodson curve (e.g. Aston-Jones & Cohen *Annu. Rev. Neurosci.* 2005), as well as recent literature on potential neural correlates of the Yerkes-Dodson curve (see point 2).

First of all, the term 'attention' is ambiguous, it usually refers to selective attention but can also mean arousal. The authors use the term attention to refer to selective attention, but it would be good to make this explicit. Also, the word 'state' is used in different connotations, which is confusing to me. In the title, the word seems to be used to describe the focus of spatial selective attention. This might not be the most suitable, as previous results from the same lab have shown that the location of the attentional focus has been shown to move around continuously, while I would say that the term 'state' generally refers to somethings that is more sustained in time. Throughout the paper, the term is mostly used to refer to brain states that are independent of the decoded spatial focus of attention, for example to arousal, which is more sustained in time, and therefore a more common way to use it.

Also, the text does not refer to the term arousal, except for once in the discussion, nor is a link made to literature on the Yerkes-Dodson curve. Instead of arousal, the term 'responsiveness' is used. I can imagine that the authors prefer to stay closer to the actual behavioral responses, but I think it that would be good to at least make a link with the more common term arousal. Then, the term 'distractibility' is generally used for what I would call a low-arousal state, 'in-task' for mid-arousal and 'impulsivity' for high-arousal state. In particular the use of the term distractibility is very confusing to me. First, the term is more common to relate it to the high arousal state instead of the low-arousal state: false alarms indicate a tendency to respond to distractors, so indicating a state of being more easily distracted (see e.g. Figure 2 in Aston-Jones & Cohen *Annu. Rev. Neurosci.* 2005). Also, distractibility is sometimes used to describe TA (line 382-385):

"We show that these two distinct behavioural markers, i.e., distractibility and impulsivity, are implemented in two different neuronal population functional components, the variance of which are respectively associated with either the decoded attentional spotlight relative to the target, or to the behavioural (perceptual) outcome produced by the subject."

Maybe I did not understand this sentence correctly, or was it just a mistake?

I would recommend to rephrase in more common terms, or at least relate to terms used to describe similar phenomenon in this literature.

2) A body of literature has recently found physiological correlates of the Yerkes-Dodson curve, which are not referenced. Particularly relevant I would say are the paper by McGinley, David & McCormick, *Neuron* 2015, as well as Snyder, Morais, Willis & Schmidt *Nature Neuroscience* 2015. McGinley et al. described how pupil size can be used as estimate of arousal in mice, and how pupil size shows a consistent relationship with behavior (hits, misses and false alarms), as well as a number of neuronal measures. Similarly, Snyder et al. found that the amplitude of alpha oscillations in monkey visual cortex shows a U-shape relationship with behavior (response times), as well as with noise correlations. Both of these papers are not referenced, nor is a link provided with either pupil size or rhythmic activity. A possible link with the locus coeruleus is mentioned, but without reference to pupil diameter, which is commonly made in this context. It would be relevant to see how the 2 components the last dPCA analysis (dPC1 and dPC2, see Figure 8), relates to the different physiological measures that have been shown to be related to the Yerkes-Dodson curve, in particular how pupil size and oscillatory activity are related to dPC2 and noise correlation to dPC1.

It seems that pupil diameter was not recorded, but oscillatory activity has been reported in previous papers by the same group. Maybe the authors would prefer to leave that out of the paper, as it is already part of the manuscript by Guillard et al bioRxiv? In any case, it would be important to reference this closely related literature.

The current paper also reports the commonly found U-shaped relationship between behavior on the one hand and amplitude of the stimulus evoked response or noise correlations on the other hand (see also e.g. Gutnisky, Beaman, Lew & Dragoi eLife 2017). However, no link is provided with the dPC analysis. Would it be possible to see how dPC2 (as estimated in the pre-target period) relates to stimulus evoked response and noise correlation (in the post-target period)? This would make the story more coherent, as well as make it easier to relate to previous literature.

Minor comments:

- On line 49-51 it is written: "This has a high relevance to the understanding of the neural bases of attention disorders such as attention deficit and hyperactivity disorder (ADHD), as recent studies describe a dissociation between spatial attention deficits and impulsivity in ADHD patients (Roberts et al., 2018)." It doesn't seem this paper has measured impulsivity, they merely found that spatial attention is not impaired in ADHD, potentially implying deficits in other cognitive functions.
- The sentence on line 52-59 is very difficult to follow.
- When describing the dPCA in the introduction (line 87 and onwards), it would be useful to already make explicit that different trials were used to decode attentional focus and dPCA.
- The sentence on line 99-101 is hard to understand for readers not familiar with the Locus Coeruleus. It would be good to give a bit more context.
- Panel A of Figure 1 is difficult to understand. The cue is very hard to see. And wouldn't you expect attention to already start to shift during the presentation of the cue? Also, dashed lines are used both for eye position and attentional focus. Might be good to choose different ways to indicate them (e.g. different color).
- In panel B of Figure 1, it's not very clear that FEF in two hemispheres was targeted. Maybe choose a view from the side or the top of the brain?
- In Figure 4A, it would be interesting to see the two neurons highlighted that correspond to panel 3AB.
- I assume 85.7 on line 355 is in degrees? Would be relevant to add that.
- Would it be possible to add to figure 8 a scatterplot, as in Figure 5F, to visualize the distribution of the two PCs in the population?
- On line 372-373 it is written: "Likewise, none of the results reported in this section changed when performed on hit, miss and false alarm trials that were equalized for TA metrics on the session level." However, no results are shown for this, which would be useful, even if the main results would stay the same. Or just leave out this sentence.
- Figure 9 is vague and confusing to me, not only because of the somewhat unusual terms (see above), but also because it mixes terms used to describe behavior or cognitive functions with terms used to describe neural measures. For example, 'responsiveness' can apply both to behavior and neural measures. Is that on purpose? I'm not sure that being ambiguous is helpful for the readability of the

manuscript. Also, there seems to be no label on the y-axis in panel b. In previous literature, it is often labeled as performance, but then the U-shape is inverted.

- It would be relevant to know whether the same dataset has already been used in a previous study, to be able to better relate it to previous literature.

- There are some spelling mistakes, for example the word 'of' should probably be removed in line 48 (in "propensity of to"), as well as the capital letter F on line 696 (in "Ffalse").

REVIEWER COMMENTS

We would like to thank the editor and the reviewers for their appreciation of our work and their constructive feedback on the manuscript. By taking into account this feedback, our revised manuscript is now strengthened and clarified. All comments have been addressed as described below (blue responses, changes to main text in bold blue) and correspondingly modified in the main text. In the process of addressing the reviewers' concerns, we detected an error in our scripts that we used for the analysis of the results corresponding to the figures 5 and 6. In the original analysis, our trial selection was biased, i.e., for one condition we selected the trials at the beginning of the session while the rest of conditions contained trials selected along the entire session. We have now repeated our analysis selecting correctly the trials for all conditions. Qualitatively, we did not find differences with our previous results. However, we have updated figures 5 and 6 accordingly. We apologize for this mistake.

New figures 5 and 6 are now reproduced below.

Figure 5:

Figure 6:

Reviewer #1 (Remarks to the Author):

The authors have studied the neural basis of spatial attention in prefrontal neurons. The authors show that in addition to encoding spatial attention, prefrontal neurons also encode a distractibility-to-impulsivity state. The authors postulate that the pre-existing neuronal states describing task-independent behavioral states sheds new light on attention disorders such as attention deficit-hyperactivity disorder.

The experiments are straightforward, the manuscript is written clearly and should clearly be considered for publication in this journal. The conclusions are reasonable. However, I have some questions regarding the methodological treatment of the data.

Major comment:

1- The authors focused their analysis on the epoch immediately preceding the target or distractor presentation ([-300 0] ms with respect to target or distractor onset). Therefore, trial history and the role of the preceding trial could be critical. Would it be possible to relate their finding to possible activity in trial $t-1$?

We thank to this reviewer for this relevant comment. To address this question, we conduct the following analysis. For each session, we extract the two dPC components associated to hit, miss and false alarm. Each trial (hit, miss or false alarm) is projected onto these two components, and we compute the correlation between the projected firing rates on any given trial with the projected firing rate of the previous trial (irrespective of its label) onto the same component.

We find that the projected activity of the current trial onto each of the two demixed principal components highly correlates with the projected activity onto the previous trial (irrespective of the trial type of the previous trial). We found that this effect is consistent across sessions. We have now included a supplementary figure S8 with this analysis.

This supplementary figure is now cited in the main text as follows: **“In addition, we tested whether the firing rates projected in these two dPC (per each trial type) were linked to the activity obtained in the previous trial (irrespective of the trial type of the preceding trial). In this context, we found that the firing rate projected in the behavior-related components correlated with the activity of the preceding trial projected in the same space (Supplementary figure S8). These results indicate that the demixed components identified based on behavioral outcome can be identified irrespective of specific task-related processes and describe a neuronal process that extends across trials. This nicely dovetails with previous findings suggesting a low-frequency dynamics of the activity of a state linked to the sensory processing efficiency, encompassing multiple trials over temporal scales of several minutes ((Amengual and Ben Hamed, 2021; Gaillard et al., 2021))**

This supplementary figure and its legend are duplicated below:

Supplementary Figure S8. (A) Scatter plot representing the distribution of the pre-cue firing rate projected onto dPC1 (U-shape) and dPC2 (Linear) of the current trial (for hit trials, miss trials and false alarm trials) with the projected firing rates obtained from the previous trial (irrespective of the trial type) for one representative session. (B) Spearman correlation coefficients of the same correlations performed in (A) for all sessions, per trial type (current trial, hit (blue), miss (red) and false alarm (yellow) responses; squares, $p < 0.05$; circles, $p > 0.05$).

2- In his paper Brody have shown that two biologically plausible ways of departing from independence can generate peaks very similar to spike timing peaks are described: covariations over trials in response latency and covariations over trials in neuronal excitability. Could the authors state over which plausible causation they observed some noise correlation in FEF. Brody CD. Correlations without synchrony. Neural Comput. 1999 Oct 1;11(7):1537-51. doi: 10.1162/089976699300016133. PMID: 10490937.

We thank to the reviewer for this comment on the possible functional origin of noise correlation. As suggested by Brody et al. (1999), spike timing synchronization (measured by the noise correlation) might arise from independent mechanisms such as covariation in neuronal excitability, or covariations in the latency of the neuronal response. The question is thus to what extend the differences in noise correlation as a function of the trial type (hit trial, miss trials or false alarm trials) that we report here are explained by fluctuations in the response latency or by fluctuations in neuronal excitability. In our study we report that the difference in noise correlation as a function of trial type arises both in the cue to target interval as well as in the pre-cue period, therefore, in a time-period where spiking activity is not modulated by any common incoming stimulus onset. As a result, we propose that the reported differences in noise correlations result from covariations in the excitability of the neurons, possibly due to more global variations in noradrenergic (Dragone et al., 2018);(Reynaud et al., 2019)) or cholinergic ((Everitt and Robbins, 1997; McCormick, 1992; McGinley et al., 2015a; Yüzgeç et al.,

2018)Everitt and Robbins, 1997; McCormick, 1992; McGinley et al., 2015; Yüzgeç et al., 2018) neuromodulation.

This point is now introduced in the result section as follows: **“Given that these reported differences in noise correlation as a function of trial type arise both during the cue to target interval as well as in the pre-cue period, we propose that they result from covariations in the excitability of the neurons, possibly due to more global variations in noradrenergic (Dragone et al., 2018) or cholinergic (Everitt and Robbins, 1997; McCormick, 1992; McGinley et al., 2015; Yüzgeç et al., 2018) neuromodulation, rather than to covariations in response latency (Brody, 1999).”**

Reviewer #2 (Remarks to the Author):

The manuscript builds on top of an impressive body of literature from the same lab, using decoding methods of neuronal activity recorded in the frontal eye fields of macaque monkeys to get an estimate of the attentional focus of the subjects. Observing that this estimate cannot explain all variability in the data, the authors use the behavioral response made by the subject (hits, misses and false alarms) to decode the brain state of the subjects. They use a demixed principal component analysis (dPCA) to investigate these two components (attention and behavioral response) in more detail and find that they are partly overlapping and partly orthogonal. Interestingly, the distance of the attentional focus relative to the target location ('Target-Attention distance', or TA) is negatively correlated with the overlap between the two components. In a separate dPCA they analyze the behavioral components in more detail and report two main components (dPC1 and dPC2), one that seems to reflect (the inverse of) arousal and the other that corresponds to the effect of arousal on performance, a U-shape with an optimum at mid-levels of arousal (similar to the Yerkes-Dodson curve).

This is a highly original work, using state-of-the-art recording techniques and analysis methods to disentangle the different neuronal population dynamics that determine performance on an attentional task. The methods are generally sound, and the figures are clear. I therefore strongly support publication of it.

At the same time, some aspects of the paper can be improved, in particular the readability and the relationship to previous literature on the topic.

General comments:

1) The text is sometimes a bit convoluted, and the terms used to interpret the results are not always clear. Related to that, reference to some of what seems to me like the most relevant literature is lacking, in particular the classic Yerkes-Dodson curve (e.g. Aston-Jones & Cohen Annu. Rev. Neurosci. 2005), as well as recent literature on potential neural correlates of the Yerkes-Dodson curve (see point 2).

We take this point of reviewer #2 very positively. We have indeed struggled with assigning the appropriate nomenclature to the different identified neural signatures. As suggested below by reviewer #2, we have tried to remain as close as possible to the behavioural responses and not be over speculative. It appears that this has come at the cost of clarity and we apologize for that. We have now tried to address this issue all throughout the manuscript, where appropriate (see below). In addition, we have now extended our discussion of the Aston-Jones review of 2005 and added the recent literature suggested by reviewer #2 (see below).

First of all, the term 'attention' is ambiguous, it usually refers to selective attention but can also mean arousal. The authors use the term attention to refer to selective attention, but it would be good to make this explicit. Also, the word 'state' is used in different connotations, which is confusing to me. In

the title, the word seems to be used to describe the focus of spatial selective attention. This might not be the most suitable, as previous results from the same lab have shown that the location of the attentional focus has been shown to move around continuously, while I would say that the term 'state' generally refers to somethings that is more sustained in time. Throughout the paper, the term is mostly used to refer to brain states that are independent of the decoded spatial focus of attention, for example to arousal, which is more sustained in time, and therefore a more common way to use it.

We absolutely agree with reviewer #2 on the necessary distinction between spatial attention which we have shown to be highly dynamic and more sustained states as we describe here and we apologize for the confusion. We have now implemented the following changes in the manuscript:

The title has been changed as follows: **“Distractibility and impulsivity neural states are distinct from selective attention and modulate the implementation of spatial attention”**

All throughout the manuscript, we use the decoding of selective attention to obtain to access to the attentional spotlight. We then compute our parameter of interest (Attention-to-target distance, TA), which represents how close the decoded attentional spotlight is to the actual expected position of the target, irrespective of the target position. This parameter thus captures how well the spatial attention is focussed on the cued location, rather than where in space. We thus now consistently refer to TA as the **focus of attention** all throughout the manuscript.

Last, we acknowledge that our use of the term **“state”** was not consistent in the manuscript, and we should only use this term when we refer to the components representing the variance associated with overt behaviour. Indeed, we show, in response to reviewer #1, a history effect on the firing rates projected onto the overt-behaviour related components, which reinforces the idea that such activity reflects a sustained effect in time and, therefore, a state of activity. Accordingly, we have changed this all throughout the manuscript.

Also, the text does not refer to the term arousal, except for once in the discussion, nor is a link made to literature on the Yerkes-Dodson curve. Instead of arousal, the term 'responsiveness' is used. I can imagine that the authors prefer to stay closer to the actual behavioral responses, but I think it that would be good to at least make a link with the more common term arousal. Then, the term 'distractibility' is generally used for what I would call a low-arousal state, 'in-task' for mid-arousal and 'impulsivity' for high-arousal state. In particular, the use of the term distractibility is very confusing to me. First, the term is more common to relate it to the high arousal state instead of the low-arousal state: false alarms indicate a tendency to respond to distractors, so indicating a state of being more easily distracted (see e.g. Figure 2 in Aston-Jones & Cohen Annu. Rev. Neurosci. 2005).

We agree with this comment from reviewer #2. It is true that we kept away from using the term “arousal” when interpreting the functional meaning of the components extracted with the dPCA, and instead we used the term “responsiveness”. We believe that the latter is more straightforward than the former, since it relates specifically to the monkey’s behaviour spanning from paucity in behaviour (miss trials) to impulsivity (false alarm trials). While there is most probably a link between this responsiveness and arousal levels, novel experiments will be necessary to attribute this type of variance to different levels in arousal, understood as a broader mental state associated with different levels of consciousness, alertness and information processing. We have now proceeded to the following changes in the main manuscript:

First, we have changed the subsection title in the results section from *Prefrontal cortex codes for behavioral distractibility and impulsivity* to ***Prefrontal cortex encodes for different levels of behavioral responsiveness.***

We have refined our definition of distractibility in the introduction as follows: “However, little is known on how such internal states are organized at the neuronal population level, how they functionally interact with attentional processes and how they account for different patterns of behavioral performance characterized by either distractibility (**i.e., the subject’s inattention to both task-relevant and task-irrelevant items**) or impulsivity (i.e., the subjects’ propensity to respond to incoming stimuli irrespective of their relevance).” And: “One of the state’s dimension defines a continuum of coding between distractibility (**absence of response**), optimal response and impulsivity (**inappropriate response**), possibly suggesting an association with proactive inhibition and response threshold adjustment (**Cowley et al., 2020**).” This definition has also been emphasized at different locations in the manuscript including in Figure 9.

In addition, following on reviewer #2’s suggestion, we introduce a new discussion paragraph in which we describe that responsiveness levels might be linked to fluctuations in arousal and wake states. This new discussion paragraph reads as follows:

“... We would thus like to propose that this component is associated with the level of optimal (in-task) behavioural regime as described in the Locus Coeruleus noradrenergic neurons (Aston-Jones and Cohen, 2005).

Prior studies have argued that the different levels of the NE system might mediate arousal states (Berridge and Waterhouse, 2003; Robinson, 1993). However, other studies refine the understanding of the LC-NE function by relating it with optimization of reward-seeking behavior (Aston-Jones and Cohen, 2005). Indeed, although the term arousal is broadly used to describe a mental state associated with different levels of consciousness and alertness, it has proven difficult to link it to specific neurobiological mechanisms (Aston-Jones and Cohen, 2005). Therefore, we would like to remain cautious when trying to associate our findings with different levels of arousal, in the absence of more direct causal evidence (see below). Indeed, the causality of the relationship between the FEF U-shaped functional component that we report in our data and LC-NE tonic discharge rate that we suggest here is very indirect at this stage and will have to be tested experimentally.”

Also, distractibility is sometimes used to describe TA (line 382-385): “We show that these two distinct behavioural markers, i.e., distractibility and impulsivity, are implemented in two different neuronal population functional components, the variance of which are respectively associated with either the decoded attentional spotlight relative to the target, or to the behavioural (perceptual) outcome produced by the subject.” Maybe I did not understand this sentence correctly, or was it just a mistake? I would recommend to rephrase in more common terms, or at least relate to terms used to describe similar phenomenon in this literature.

We thank reviewer #2 for spotting this mistake. We have changed this sentence as follows: “We show that these two distinct behavioural markers, i.e., **optimality and responsiveness, are implemented in two different neuronal population functional components, the variance of which is only partially dependent with the variance associated with the decoded attentional spotlight relative to the target**”.

2) A body of literature has recently found physiological correlates of the Yerkes-Dodson curve, which are not referenced. Particularly relevant I would say are the paper by McGinley, David & McCormick, Neuron 2015, as well as Snyder, Morais, Willis & Schmidt Nature Neuroscience 2015. McGinley et al. described how pupil size can be used as estimate of arousal in mice, and how pupil size shows a consistent relationship with behavior (hits, misses and false alarms), as well as a number of neuronal measures. Similarly, Snyder et al. found that the amplitude of alpha oscillations in monkey visual cortex shows a U-shape relationship with behavior (response times), as well as with noise correlations.

Both of these papers are not referenced, nor is a link provided with either pupil size or rhythmic activity. A possible link with the locus coeruleus is mentioned, but without reference to pupil diameter, which is commonly made in this context. It would be relevant to see how the 2 components the last dPCA analysis (dPC1 and dPC2, see Figure 8), relates to the different physiological measures that have been shown to be related to the Yerkes-Dodson curve, in particular how pupil size and oscillatory activity are related to dPC2 and noise correlation to dPC1.

We appreciate this comment from this reviewer. We have now implemented a novel section in the discussion of the paper addressing this concern. Also, see our response to the next point of reviewer #2. The new discussion reads as follows:

“Physiological correlates of the behavior-related components

In the previous section we have argued that the different levels of activation of the neural states associated with optimality and responsiveness contribute to the processing of sensory information and its behavioral consequences. However, there is a lack of consensus on which physiological mechanisms might correlate with such state changes and account for either an optimal or sub-optimal sensory encoding or behavioral performance. Prior studies have shown that pupil size might be a potential candidate as a proxy of these fluctuations in state, as it has been shown that it impacts on both the sensory evoked response, on the spontaneous activity of cortical responses (McGinley et al., 2015b), as well as on membrane potential level (McGinley et al., 2015a). Indeed, McGinley and colleagues (2015a) have shown, in mice, that optimal signal detection behavior and evoked responses occur at intermediate arousal (measures by pupillometry) when pre-decision membrane potentials are hyperpolarized, revealing a cortical physiological signature of the inverted-U relationship between performance and arousal. Relating to these findings, we have recently described consistent low-frequency fluctuations in the pupil size from human data recorded during the execution of an exploration task, in the order of a few cycles per hour. These rhythmic fluctuations are linked with different behavioral states characterized by differences in detection times as well as in prefrontal attention and perception information capacity (Gaillard et al, 2021) and, possibly, the activity of the noradrenergic system (Reynaud et al., 2019).

Another possible physiological correlate of the activity reflecting these neural states could be the global network influence on the local functional connectivity. In this context, Snyder and colleagues (2015) showed that spike count correlation in area V4 of the macaque correlated with certain properties of the EEG, and specifically with the level of alpha-band oscillatory activity measured in different regions of the occipital cortex, as well as with the reaction times during a spatial attention task. Relevant to the present discussion, this relationship between the EEG alpha oscillation and the spike count correlation and behavior were U-shaped. Other studies have shown the same inverted U-shape relationship between EEG oscillations during attention-demanding detection tasks and performance (Ai and Ro, 2014; Lange et al., 2012; Linkenkaer-Hansen, 2004).

All in all, our results are in line with previous studies showing a non-monotonic relationship between different levels of activity linked with brain state, measured with different physiological measurements and specific behavioral regimes.”

It seems that pupil diameter was not recorded, but oscillatory activity has been reported in previous papers by the same group. Maybe the authors would prefer to leave that out of the paper, as it is already part of the manuscript by Gaillard et al bioRxiv? In any case, it would be important to reference this closely related literature.

Reviewer #2 is correct. Unfortunately, at the time when this data was collected we did not record pupil size data, and therefore these analyses cannot be addressed. Currently, we are running different experiments in monkey and human populations where we are recording the pupil size in addition to electrophysiological data. In the reference that the reviewer cites (Gaillard et al, 2021), pupil size data is available for the human sample, not for the monkeys. However, the intuition of reviewer #2 is correct: a set of preliminary analyses suggests a functional link between the states described in the present work and the ultra-slow rhythmic changes in attention and perception described in Gaillard et al. 2021. This results have yet to be consolidated in order to propose a unified interpretational framework for these observations.

The current paper also reports the commonly found U-shaped relationship between behaviour on the one hand and amplitude of the stimulus evoked response or noise correlations on the other hand (see also e.g. Gutnisky, Beaman, Lew & Dragoi eLife 2017). However, no link is provided with the dPC analysis. Would it be possible to see how dPC2 (as estimated in the pre-target period) relates to stimulus evoked response and noise correlation (in the post-target period)? This would make the story more coherent, as well as make it easier to relate to previous literature.

We thank reviewer #2 for this very relevant comment. In order to provide a functional link between the dPC1 (U-shape relationship between trial type estimated in the pre-target period) and noise correlation, we have now removed the variance explained by the U-shape demixed component by projecting the pre-target activity contained in the rest of the components back onto the original data space, and we have re-calculated the noise correlation in each of the trial types. As a result, we observed that the U-shape relationship in the noise correlation as a function of the trial type vanishes (supplementary figure S6a). We repeated the same analysis but removing the linear component instead of the U-shape component. When recomputing noise correlations from the back projected activity, we obtain the same U-shape relationship in the noise correlation as a function of the trial type as described on the original data (supplementary figure S6a). Therefore, we interpret this finding as a proof of a functional link between the activity reflected in the dpc1 and the noise correlation.

Regarding the relationship between the 2-dimensional neural state that we observe in the pre-target period and the stimulus-evoked response, we proceed similarly to what is described above: We removed the variance associated with either the dPC2 (linear component) or the dPC1 (U-shaped component) estimated in the pre-target period and we projected the stimulus evoked response back onto the original dataset space. We observe that the firing rate differences during the time interval corresponding to the evoked response between miss trials and false alarm trials vanishes when we remove the variance explained by the dPC2 (linear). This is mostly accounted for by a change in the pre-target activity (supplementary figure S6b). This stimulus-evoked response is maintained when we removed the variance explained by the dPC1 (U-shape component) (supplementary figure S6b). This indicates that none of the dPCs account for the evoked behavioural response.

This is now described in the results section as follows: **“A U-shaped relationship between behaviour on the one hand and amplitude of the stimulus evoked response or noise correlations on the other hand has already been described by others (Gutnisky et al., 2017). In order to better characterize the relationship between these two identified components (U-shaped dPC1 and linear dPC2 respectively) and noise correlation, we measured noise correlation levels in the three different trial types after removing the variance explained by one of the two components and projecting the pre-target activity contained in the rest of the components back onto the original data space. The differences in noise correlation between trial types vanishes when removing the variance associated with dPC1 but not when removing the variance associated with dPC2 (supplemental figure S6a). This result indicates a functional link between noise correlation levels and the activity represented by**

the dPC1. Likewise, in order to better characterize the relationship between these two identified components (U-shaped dPC1 and linear dPC2 respectively) and stimulus-evoked response, we removed the variance associated with either the dPC2 (linear component) or the dPC1 (U-shaped component) estimated in the pre-target period and we projected the stimulus evoked response back onto the original dataset space (supplemental figure S6b). While firing rates extracted from miss trials and false alarm trials in post target time interval (corresponding to the stimulus evoked responses) after removing the variance explained by the dPC1 did not differ relative to the original neuronal response, we observe that the firing rate differences during the time interval corresponding to the evoked response between miss trials and false alarm trials vanishes when we remove the variance explained by the dPC2. This is mostly accounted for by a change in the pre-target activity (supplementary figure S6b). This indicates that none of the two dPCs account for the evoked response to the target but that dPC1 specifically accounts for pre-target differences in noise correlations and dPC2 specifically accounts for pre-target differences in average firing rate.”

This is also introduced in the discussion as follows: “Overall, this points to two independent neuronal processes contributing to behavioral outcome in addition to the classically-described attentional orientation (Figure 9B). **None of the two components account for the evoked response to the target. However, the U-shaped component specifically accounts for pre-target differences in noise correlations and the linear component specifically accounts for pre-target differences in average firing rate.**” And: “The U-shape observed in the projection onto the first behavioral outcome component of FEF neuronal population firing rates **accounts for** the U-shape observed for noise-correlation levels associated to the same behavioral conditions (Figure 9B, **supplemental figure S6A).**”

Supplementary figure S6 and its legend are appended below:

Figure S6 A. (Up) Boxplot of noise correlations across sessions, during a pre-target interval (-300 to 0 ms), for each of the three trial types: Hit (blue), Miss (green) and false alarm trials (red). (Bottom) Boxplot of noise correlations across sessions from data reconstructed after removing the variance explained by dPC1 (U-shape, Left), or dPC2 (Linear, Right). Wilcoxon test between trial types were performed (* p < 0.05, ** p < 0.01). B. (Up) Mean firing rates recorded during Hit (blue), Miss (green) and false alarm trials (red) locked to target onset (interval -400 to 400 ms). (Bottom). Mean firing rates averaged across similar trial types, after removing the variance explained by dPC1 (U-shape, Left), or dPC2 (Linear, Right).

In addition, the following sentence has been added to the discussion: **“This is in line with yet other studies which have identified different states in the population activity associated with differences in evoked responses and correlated variability (Gutnisky et al., 2017)”**.

Minor comments:

- On line 49-51 it is written: “This has a high relevance to the understanding of the neural bases of attention disorders such as attention deficit and hyperactivity disorder (ADHD), as recent studies describe a dissociation between spatial attention deficits and impulsivity in ADHD patients (Roberts et al., 2018).” It doesn’t seem this paper has measured impulsivity, they merely found that spatial attention is not impaired in ADHD, potentially implying deficits in other cognitive functions.

We have changed this sentence as follows: **“... as recent studies describe that allocation of attention is not impaired in these patients (Roberts et al., 2018), and therefore other processes presumably related with the internal states described here could be affected.”**

- The sentence on line 52-59 is very difficult to follow.

We have change this sentence as follows: **“The question we address in the present work is whether behavioral performance during an attentional task is (non-exclusively) (i) a direct consequence of a good or a miss allocation of spatial attention relative to task events (Astrand et al., 2020; Buschman and Miller, 2007; Di Bello et al., 2021; Ibos et al., 2013; Moore and Armstrong, 2003; Thompson and Schall, 2000, 1999), or (ii) whether behavioral performance is determined by underlying internal states that can be precisely assigned to specific functional neuronal states of the prefrontal cortex neural population responses (Cowley et al., 2020).”**

- When describing the dPCA in the introduction (line 87 and onwards), it would be useful to already make explicit that different trials were used to decode attentional focus and dPCA.

We have addressed this comment as follows: **“To prove our prediction, we used demixed principal component analysis (dPCA, Kobak et al., 2016; Machens, 2010), a dimensionality reduction technique that allows associating the variability explained by each component with specific task- or behavior-related parameters. We thus divided our trials based on the position of the decoded attentional spotlight with respect of the target position (target-to-attention distance, TA; Small TA, Medium TA, Large TA) and the behavioral performance (hit trials, miss trials and false alarm trials), and we extracted demixed components the variance of which was associated with either to TA or behavioural performance.”**

- The sentence on line 99-101 is hard to understand for readers not familiar with the Locus Coeruleus. It would be good to give a bit more context.

We have added the following lines: **“Consistent with this finding, prior literature has reported evidence suggesting that such behavioral responses might be linked with different activity regimes described in the Locus Coeruleus (Aston-Jones and Cohen, 2005). Therefore, activity in this dimension might correspond to the neural signature of the activity of the noradrenergic system.”**

- Panel A of Figure 1 is difficult to understand. The cue is very hard to see. And wouldn’t you expect attention to already start to shift during the presentation of the cue? Also, dashed lines are used both for eye position and attentional focus. Might be good to choose different ways to indicate them (e.g. different color).

We have adapted figure 1 following the suggestions of the reviewer. New figure 1 is now pasted below.

- In panel B of Figure 1, it's not very clear that FEF in two hemispheres was targeted. Maybe choose a view from the side or the top of the brain?

We appreciate this observation. We have now changed panel B hoping it is clearer to the reader. New figure 1 is now pasted below:

Figure 1. Methods. (A) 100 % validity cued target-detection task with distractors. To initiate the trial, monkeys had to hold a bar with the hand and fixate their gaze on a central cross on the screen. Monkeys received liquid reward for releasing the bar 150-750 ms after target presentation onset. Target location was indicated by a cue (green square, second screen). Monkeys had to ignore any un-cued event (distractors). **Monkeys were instructed to keep their gaze on the fixation point (white dashed lines), therefore they had to detect the stimuli using selective spatial attention (red dashed lines).** (B) On each session, one 24-contact recording probe was placed in **right FEF (top) and left FEF (bottom)**. (C) Single MUA mean (\pm s.e.) associated to when cue is orienting towards the preferred (black) or the anti-preferred (gray) spatial location, during the cue-to-target interval. (D) Distribution of attention modulation index (Preferred - Anti-preferred)/(Preferred + Anti-Preferred), computed over 200 ms before target onset across all MUAs of all sessions. Black histogram corresponds to channels in which the neuronal activity during this time interval was significantly different between the preferred and the anti-preferred spatial attention responses (Wilcoxon test, $p < 0.05$: black, significant difference).

- In Figure 4A, it would be interesting to see the two neurons highlighted that correspond to panel 3AB.

We thanks to the reviewer for this comment. We have now highlighted the two neurons which are classified as mixed selectivity neurons in figure 3AB in figure 4A. Figure 4A and its new figure caption are now reproduced below:

(A) Scatter plot showing attentional (x-axis) and behavioral outcome (y-axis) modulation indices (absolute value) for all recorded neurons during the time interval -400 to -100 ms before target onset. Neurons are classified based on the significant TA (red), behavioral outcome (blue) or mixed (green) tuning. **Neurons corresponding to figures 3A and 3B are indicated by larger symbols.** Pie chart shows the proportion of neurons for each type of selectivity.

- I assume 85.7 on line 355 is in degrees? Would be relevant to add that.

We agree with the reviewer that the missing units were degrees. We have fixed this mistake.

- Would it be possible to add to figure 8 a scatterplot, as in Figure 5F, to visualize the distribution of the two PCs in the population?

We now add this information as a supplementary figure S5, reproduced below, and cited in the main text as follows:

“The angle between these two components was 85.7 degrees (IQR = 6.33), and did not pass the test of non-orthogonality (see Kobak et al., 2016a). **Importantly, both components were equally distributed across the whole neuronal population, and the weights of each component showed a clear unimodal distribution centred close to zero (dPC1 component, median = 0, IQR: 0.02; dPC2 component, median = 0.01, IQR: 0.031).** This latter observation rules out the possibility that components might be exhibited only by a subset of cells (supplemental figure S5).”

Figure S5. For each neuron, we use the first (U-shape) and second (Linear) demixed PCs described in figure 8 to plot its location on the plane defined by these two components. These components present a weight distribution that tends to be centered and equally distributed around zero (cf. respective histograms). The scatterplot shows the relationship between the neurons' weights in the dPC1 and dPC2 demixed components. This correlation is non-significant ($p=0.6$). The dot product between these components indicates that these components are orthogonal (89 degrees).

- On line 372-373 it is written: "Likewise, none of the results reported in this section changed when performed on hit, miss and false alarm trials that were equalized for TA metrics on the session level." However, no results are shown for this, which would be useful, even if the main results would stay the same. Or just leave out this sentence.

We appreciate this comment. We have analysed the noise correlation per each trial type equalizing for TA metrics, and we did not find that TA affected this measure. We would like to keep this as a result of the paper, therefore we added it as a supplementary figure S4, reproduced below and cited in the main text as follows: "Likewise, none of the results reported in this section changed when performed on hit, miss and false alarm trials that were equalized for TA metrics on the session level (Supplemental figure S4)."

Figure S4. Boxplot representing the distribution of noise correlation across sessions for hit trials (blue), miss trial (green) and false alarm trials (red), after equalizing the mean TA value between trial types (Wilcoxon test, $** p < 0.01$).

- Figure 9 is vague and confusing to me, not only because of the somewhat unusual terms (see above), but also because it mixes terms used to describe behavior or cognitive functions with terms used to describe neural measures. For example, 'responsiveness' can apply both to behavior and neural

measures. Is that on purpose? I'm not sure that being ambiguous is helpful for the readability of the manuscript. Also, there seems to be no label on the y-axis in panel b. In previous literature, it is often labeled as performance, but then the U-shape is inverted.

We thank to the reviewer for this comment. Following your suggestions, we have clarified the terms along the text, and we have argued on the selection of the terminology to associate each neurophysiological pattern to a specific cognitive function. We have also changed the labels on figure 9. We believe that the figure is now more clear.

New figure 9 is now reproduced below:

Figure 9: Schema of the neurophysiological underpinnings explaining the relationship between covert attention and overt behavior. (A) Behavioral gain produced by the good allocation of attention with respect to expected target position varies as a function of the level of overlap between the functional population associated to covert attention (blank ellipses) and overt behavior (textured ellipses). (B) This latter component is associated with two pre-existing neuronal states describing two task-independent behavioural states, reflecting the degree of distractibility (no response)-to-impulsivity (inappropriate response) or responsiveness of the subject (linear) as well as degree of behavioral optimality in the task (U-shaped).

- It would be relevant to know whether the same dataset has already been used in a previous study, to be able to better relate it to previous literature.

We thank the reviewer for this comment. Indeed, the dataset has been used in a couple of previous studies. This has been addressed in the current version of the manuscript as follows: "The majority of neurons are significantly modulated depending on whether attention is oriented towards their preferred or non-preferred receptive field (Figure 1D). **This dataset has been used in prior studies from our lab (De Sousa et al., 2021; Di Bello et al., 2021; Gaillard et al., 2020)**".

- There are some spelling mistakes, for example the word 'of' should probably be removed in line 48 (in "propensity of to"), as well as the capital letter F on line 696 (in "Ffalse").

We have now thoroughly reread the manuscript and hopefully corrected all such typos. Thanks to reviewer 2 for pointing to these.

References de response to the reviewers:

- Ai, L., Ro, T., 2014. The phase of prestimulus alpha oscillations affects tactile perception. *J. Neurophysiol.* 111, 1300–1307. <https://doi.org/10.1152/jn.00125.2013>
- Amengual, J.L., Ben Hamed, S., 2021. Revisiting Persistent Neuronal Activity During Covert Spatial Attention. *Front. Neural Circuits* 15, 679796. <https://doi.org/10.3389/fncir.2021.679796>
- Aston-Jones, G., Cohen, J.D., 2005. AN INTEGRATIVE THEORY OF LOCUS COERULEUS-NOREPINEPHRINE FUNCTION: Adaptive Gain and Optimal Performance. *Annu. Rev. Neurosci.* 28, 403–450. <https://doi.org/10.1146/annurev.neuro.28.061604.135709>

- Astrand, E., Wardak, C., Ben Hamed, S., 2020. Neuronal population correlates of target selection and distractor filtering. *NeuroImage* 209, 116517. <https://doi.org/10.1016/j.neuroimage.2020.116517>
- Berridge, C.W., Waterhouse, B.D., 2003. The locus coeruleus-noradrenergic system: modulation of behavioral state and state-dependent cognitive processes. *Brain Res. Brain Res. Rev.* 42, 33–84. [https://doi.org/10.1016/s0165-0173\(03\)00143-7](https://doi.org/10.1016/s0165-0173(03)00143-7)
- Brody, C.D., 1999. Correlations Without Synchrony. *Neural Comput.* 11, 1537–1551. <https://doi.org/10.1162/089976699300016133>
- Buschman, T.J., Miller, E.K., 2007. Top-Down Versus Bottom-Up Control of Attention in the Prefrontal and Posterior Parietal Cortices. *Science* 315, 1860–1862. <https://doi.org/10.1126/science.1138071>
- Cowley, B.R., Snyder, A.C., Acar, K., Williamson, R.C., Yu, B.M., Smith, M.A., 2020. Slow Drift of Neural Activity as a Signature of Impulsivity in Macaque Visual and Prefrontal Cortex. *Neuron* 108, 551–567.e8. <https://doi.org/10.1016/j.neuron.2020.07.021>
- De Sousa, C., Gaillard, C., Di Bello, C., Ben Hadj Hassen, F., Ben Hamed, S., 2021. Behavioral validation of novel high resolution attention decoding method from multi-units & local field potentials. *NeuroImage* 231, 117853. <https://doi.org/10.1016/j.neuroimage.2021.117853>
- Di Bello, F., Ben Hadj Hassen, S., Astrand, E., Ben Hamed, S., 2021. Prefrontal Control of Proactive and Reactive Mechanisms of Visual Suppression. *Cereb. Cortex* bhab378. <https://doi.org/10.1093/cercor/bhab378>
- Dragone, A., Lasaponara, S., Pinto, M., Rotondaro, F., De Luca, M., Doricchi, F., 2018. Expectancy modulates pupil size during endogenous orienting of spatial attention. *Cortex* 102, 57–66. <https://doi.org/10.1016/j.cortex.2017.09.011>
- Everitt, B.J., Robbins, T.W., 1997. CENTRAL CHOLINERGIC SYSTEMS AND COGNITION. *Annu. Rev. Psychol.* 48, 649–684. <https://doi.org/10.1146/annurev.psych.48.1.649>
- Gaillard, C., Ben Hadj Hassen, S., Di Bello, F., Bihan-Poudec, Y., VanRullen, R., Ben Hamed, S., 2020. Prefrontal attentional saccades explore space rhythmically. *Nat. Commun.* 11, 925. <https://doi.org/10.1038/s41467-020-14649-7>
- Gaillard, C., De Sousa, C., Amengual, J., Lorient, C., Ziane, C., Hadj Hassen, S.B., Di Bello, F., Hamed, S.B., 2021. Attentional brain rhythms during prolonged cognitive activity. *bioRxiv* 2021.05.26.445730. <https://doi.org/10.1101/2021.05.26.445730>
- Gutnisky, D.A., Beaman, C., Lew, S.E., Dragoi, V., 2017. Cortical response states for enhanced sensory discrimination. *eLife* 6, e29226. <https://doi.org/10.7554/eLife.29226>
- Ibos, G., Duhamel, J.-R., Ben Hamed, S., 2013. A Functional Hierarchy within the Parietofrontal Network in Stimulus Selection and Attention Control. *J. Neurosci.* 33, 8359–8369. <https://doi.org/10.1523/JNEUROSCI.4058-12.2013>
- Lange, J., Halacz, J., van Dijk, H., Kahlbrock, N., Schnitzler, A., 2012. Fluctuations of Prestimulus Oscillatory Power Predict Subjective Perception of Tactile Simultaneity. *Cereb. Cortex* 22, 2564–2574. <https://doi.org/10.1093/cercor/bhr329>
- Linkenkaer-Hansen, K., 2004. Prestimulus Oscillations Enhance Psychophysical Performance in Humans. *J. Neurosci.* 24, 10186–10190. <https://doi.org/10.1523/JNEUROSCI.2584-04.2004>
- McCormick, D., 1992. Cellular mechanisms underlying cholinergic and noradrenergic modulation of neuronal firing mode in the cat and guinea pig dorsal lateral geniculate nucleus. *J. Neurosci.* 12, 278–289. <https://doi.org/10.1523/JNEUROSCI.12-01-00278.1992>
- McGinley, M.J., David, S.V., McCormick, D.A., 2015a. Cortical Membrane Potential Signature of Optimal States for Sensory Signal Detection. *Neuron* 87, 179–192. <https://doi.org/10.1016/j.neuron.2015.05.038>
- McGinley, M.J., Vinck, M., Reimer, J., Batista-Brito, R., Zagha, E., Cadwell, C.R., Tolias, A.S., Cardin, J.A., McCormick, D.A., 2015b. Waking State: Rapid Variations Modulate Neural and Behavioral Responses. *Neuron* 87, 1143–1161. <https://doi.org/10.1016/j.neuron.2015.09.012>

- Moore, T., Armstrong, K.M., 2003. Selective gating of visual signals by microstimulation of frontal cortex. *Nature* 421, 370–373. <https://doi.org/10.1038/nature01341>
- Reynaud, A.J., Froesel, M., Guedj, C., Ben Hadj Hassen, S., Cléry, J., Meunier, M., Ben Hamed, S., Hadj-Bouziane, F., 2019. Atomoxetine improves attentional orienting in a predictive context. *Neuropharmacology* 150, 59–69. <https://doi.org/10.1016/j.neuropharm.2019.03.012>
- Roberts, M., Ashinoff, B.K., Castellanos, F.X., Carrasco, M., 2018. When attention is intact in adults with ADHD. *Psychon. Bull. Rev.* 25, 1423–1434. <https://doi.org/10.3758/s13423-017-1407-4>
- Robinson, T., 1993. The neural basis of drug craving: An incentive-sensitization theory of addiction. *Brain Res. Rev.* 18, 247–291. [https://doi.org/10.1016/0165-0173\(93\)90013-P](https://doi.org/10.1016/0165-0173(93)90013-P)
- Thompson, K.G., Schall, J.D., 2000. Antecedents and correlates of visual detection and awareness in macaque prefrontal cortex. *Vision Res.* 40, 1523–1538. [https://doi.org/10.1016/S0042-6989\(99\)00250-3](https://doi.org/10.1016/S0042-6989(99)00250-3)
- Thompson, K.G., Schall, J.D., 1999. The detection of visual signals by macaque frontal eye field during masking. *Nat. Neurosci.* 2, 283–288. <https://doi.org/10.1038/6398>
- Yüzgeç, Ö., Prsa, M., Zimmermann, R., Huber, D., 2018. Pupil Size Coupling to Cortical States Protects the Stability of Deep Sleep via Parasympathetic Modulation. *Curr. Biol.* 28, 392-400.e3. <https://doi.org/10.1016/j.cub.2017.12.049>

REVIEWERS' COMMENTS

Reviewer #1 (Remarks to the Author):

The authors have made a delighted review of my comments. I do not have further comments, and I recommend publishing this revised manuscript.

Reviewer #2 (Remarks to the Author):

In my view, the manuscript is greatly improved, in particular the extension of the discussion and the changes in Figure 9. I have no further comments. I believe it's very creative and good work that provides an important advancement to the field.